# Mitigating Diabetic Cardiomyopathy: The Synergistic Potential of Sea Buckthorn and Metformin Explored via Bioinformatics and Chemoinformatics

**DOI:** 10.3390/biology14040361

**Published:** 2025-03-31

**Authors:** Kamran Safavi, Navid Abedpoor, Fatemeh Hajibabaie, Elina Kaviani

**Affiliations:** 1Department of Plant Biotechnology, Medicinal Plants Research Centre, Isfahan (Khorasgan) Branch, Islamic Azad University, Isfahan 8155139998, Iran; 2Department of Sports Physiology, Faculty of Sports Sciences, Isfahan (Khorasgan) Branch, Islamic Azad University, Isfahan 8155139998, Iran; 3Department of Biology, Faculty of Basic Sciences, Shahrekord Branch, Islamic Azad University, Shahrekord 8813733395, Iran; fateme.hajibabaii1991@gmail.com; 4Cancer Prevention Research Center, Isfahan University of Medical Sciences, Isfahan 8184917354, Iran; elinakaviani87@gmail.com

**Keywords:** diabetic cardiomyopathy, sea buckthorn, metformin, oncostatin, ferroptosis, chemoinformatics, bioinformatics

## Abstract

Diabetic cardiomyopathy is a serious heart condition linked to type 2 diabetes that can lead to heart failure. This study explored how a natural extract from the sea buckthorn plant, combined with the diabetes medication metformin, can help treat this condition in mice. Researchers induced type 2 diabetes in the mice through a high-fat diet and then treated them with either sea buckthorn extract, metformin, or both. The results showed that the combination therapy significantly improved blood sugar control, reduced inflammation, and protected the heart from damage. Furthermore, the combined treatment was more effective than using either option alone in preventing heart damage. These findings suggest that using sea buckthorn extract along with metformin could be a promising new strategy for treating heart problems in people with diabetes, which could greatly benefit public health by improving the management of diabetic heart disease.

## 1. Introduction

Diabetes mellitus, a chronic metabolic disorder arising from a complex interplay of genetic and environmental factors, presents a significant global health challenge due to its rising prevalence [1,2]. Recent data indicate that diabetes affects an estimated 10.5% of the worldwide population aged 20–79 years, equating to 536.6 million individuals. Furthermore, projections suggest a continued upward trajectory, with prevalence expected to reach 12.2% (783.2 million individuals) by 2045 [3]. This concerning trend underscores the urgent imperative to develop and implement effective prevention and management strategies to mitigate the substantial individual and societal burden of diabetes [4]. Individuals with diabetes experience a significantly heightened risk of cardiovascular complications, primarily attributed to chronic metabolic dysregulation, persistent hyperglycemia, and insulin resistance (IR) inherent to the disease [5]. Among these complications, diabetic cardiomyopathy (DCM) has emerged as a particular concern [2]. DCM, characterized by non-ischemic myocardial dysfunction, manifests as structural and functional abnormalities within the myocardium of diabetic individuals in the absence of traditional cardiac risk factors such as coronary artery disease, hypertension, and significant valvular disease [6].

Despite increasing research interest in DCM, a universally accepted definition remains elusive. This lack of a clear definition contributes to ongoing uncertainties regarding this complex condition’s true prevalence and incidence [7]. Early DCM features have been estimated to occur in 25–33% of asymptomatic diabetic patients [8]. Despite substantial research efforts over the past decade aimed at elucidating the pathogenesis and clinical presentation of DCM, effective preventative and therapeutic strategies remain limited [9]. The underlying mechanisms driving DCM are multifaceted and intricately linked to the dysregulated glucose and lipid metabolism inherent to diabetes mellitus. This metabolic dysregulation contributes to heightened oxidative stress and triggers a cascade of inflammatory pathways, ultimately culminating in cellular and extracellular damage within the myocardium. These pathological processes impair diastolic and systolic cardiac function, forming the foundation of DCM [10]. The pathogenesis of diabetic cardiomyopathy (DCM) is intricately linked to the metabolic derangements’ characteristic of diabetes. Hyperglycemia, a hallmark of diabetes, drives the formation and accumulation of advanced glycation end products (AGEs) within the cardiac extracellular matrix [11,12]. Concurrently, cardiomyocytes undergo a metabolic shift towards increased fatty acid utilization, resulting in intracellular lipid accumulation and lipotoxicity. AGE accumulation and intracellular lipid overload contribute to the excessive generation of reactive oxygen and nitrogen species, initiating a cascade of oxidative stress and inflammation underlying DCM development [13].

Cell death pathways are central to DCM pathogenesis, with apoptosis recognized as the predominant mode of cardiomyocyte loss in the early stages [14]. However, the contribution of non-apoptotic and non-necrotic forms of regulated cell death, such as ferroptosis, to DCM progression, particularly in later stages, remains largely unexplored. Understanding the specific roles of these alternative cell death pathways may reveal novel therapeutic targets for mitigating DCM progression [15].

Ferroptosis, a recently characterized form of regulated cell death, is defined by its dependence on iron and distinctive features, including excessive iron accumulation, rampant lipid peroxidation, and depletion of antioxidant systems, notably glutathione peroxidase 4 (GPX4) [16,17]. This iron-dependent, non-apoptotic cell death modality was termed “ferroptosis” by Dixon et al. in 2012 [18]. Ferroptosis can be initiated through diverse pathways that ultimately converge on GPX4, directly or indirectly impairing its function. This disruption of GPX4-mediated antioxidant defense leads to the unchecked accumulation of lipid reactive oxygen species (ROS), ultimately overwhelming the cellular capacity to detoxify these damaging species and culminating in cell death [19].

Non-coding RNAs (ncRNAs) have garnered significant attention recently due to their multifaceted roles in gene expression regulation. These RNA molecules, transcribed from the genome but not translated into proteins, exert their regulatory effects through intricate interactions within the genetic network, ultimately impacting various cellular processes, including differentiation, metabolism, transcription, and proliferation [20]. Growing evidence points to a critical role of ncRNAs, encompassing microRNAs (miRNAs), circular RNAs (circRNAs), and lncRNAs, in the pathogenesis of pyroptosis within the context of the diabetic heart. This highlights the potential for ncRNA-targeted therapeutic interventions for diabetic cardiomyopathy. The competitive endogenous RNA (ceRNA) theory explains how lncRNAs and circRNA function as sponging miRNAs [21]. Studies have reported the ceRNAs in the process of ferroptosis [22,23,24].

Metformin, a widely prescribed anti-diabetic medication, has been shown to attenuate hyperlipidemia-induced vascular calcification through its anti-iron effects [25]. This observation suggests that metformin’s anti-ferroptotic properties may contribute to its therapeutic benefits in managing diabetes [26,27]. Further research is warranted to confirm whether this mechanism plays a significant role in its anti-diabetic effects. Beyond metformin, numerous natural compounds have demonstrated the ability to protect cells from ferroptosis. The following section will explore how these plant-derived components modulate ferroptosis and their potential therapeutic implications for type 2 diabetes mellitus (T2DM) [28,29]. While metformin is generally well tolerated, it is essential to note that it can cause gastrointestinal side effects, particularly during the initial stages of treatment. These side effects commonly include dyspepsia, nausea, and diarrhea. Furthermore, metformin should be avoided in patients with severely compromised renal function, decompensated heart failure, severe liver disease, or other serious medical conditions [27].

Sea buckthorn (*Hippophae rhamnoides* L.) is a deciduous shrub or tree known as Siberian pineapple, sand thorn, seaberry, and sallow thorn [30]. *Hippophae* L. is native to the Hengduan Mountains and East Himalayas and widespread throughout temperate Eurasia [31]. This plant’s fruits, leaves, stems, branches, roots, and thorns have been used in medicine as a dietary supplement, to conserve soil and moisture, and to create animal habitats” [32]. Sea buckthorn is known as a “natural repository of vitamins” and “provider of nutrition and healthcare” due to its 200 valuable and bioactive components. Food manufacturers employ sea buckthorn to produce breads, yogurts, jams, beverages, teas, and more [33,34]. Traditional medicine uses sea buckthorn to treat slow digestion, stomach problems, cardiovascular disease, liver damage, skin diseases, and ulcers [35]. Sea buckthorn has been studied for its pharmacological properties, including cancer prevention, inflammation reduction, microbe and virus defense, and cardiovascular protection [36,37,38]. Due to its high vitamin, carotenoids, polyphenol, and fatty acid content, sea buckthorn has great pharmacological and therapeutic potential [39,40]. Recently, more countries have identified sea buckthorn’s therapeutic properties and created an industry around it.

This study embarks on a pioneering exploration to unravel the intricate genetic underpinnings of ferroptosis in the context of heart failure arising from T2DM. Leveraging the power of bioinformatic and in silico methodologies, we aim to pinpoint the critical hub genes orchestrating this complex interplay. Furthermore, we delve into the realm of chemoinformatic analysis to unearth promising therapeutic targets and agents, with a particular focus on metformin, paving the way for novel treatment strategies for this debilitating condition.

## 2. Materials and Methods

### 2.1. Application of System Biology Analysis

The present analysis applied network visualization approaches to examine gene expression data, designing a model that depicts the progression of DCM. The study of pathogenic mechanisms involved in the progression of DCM includes analyzing ceRNA interactions and the molecular signaling systems that are key in the occurrence of DCM pathology. The microarray profiles related to DCM were acquired from the GEO database, which may be accessed with the search phrase “diabetic cardiomyopathy” at https://www.ncbi.nlm.nih.gov/geo (accessed on 21 May 2024). The raw data from PBMC specimens in GSE156993 and heart sample in GSE26887 were analyzed using the R programming language program (V: R 4.3.2, R 4.0.2) with Bioconductor packages (V: 3.20). The GSE156993 dataset included five samples from T2DM-poorly DL-P individuals and six from healthy participants. The PBMC of patients who are afflicted by a combination of T2DM, dyslipidemia, and periodontitis exhibit changes in their molecular profiles, primarily related to the inflammatory response, immune cell movement, and pathways linked with infectious diseases. Poorly managed T2DM is defined as having a glycated hemoglobin (HbA1c) level of 8.5% or higher. On the other hand, well-controlled T2DM patients have an HbA1c level below 7.0%. The use of the MAS5 method was crucial in normalizing the data.

Moreover, the GSE26887 dataset included five samples from control subjects and seven diabetic heart failure samples. The GSE26887 dataset, included left ventricle (LV) cardiac biopsies retrieved from the key, non-infarcted region (distant zone) of patients with dilated hypokinetic post-ischemic cardiovascular disease who were undergoing surgical ventricular restoration procedures. The inclusion criteria for diabetes were: Glycemia ≥ 126 mg/dL, prior diagnosis of T2DM or anti-diabetic treatment, while for non-diabetic: Glycemia < 100 mg/dL and HbA1c: normal values 4.8–6.0%. Additionally, heart failure patients were matched based on End Systolic Volume (ESV), Left Ventricular Ejection Fraction (LVEF), age, sex, ethnic distribution, smoking behaviors, high blood pressure, glomerular filtration rate (GFR), and body mass index (BMI). Gene expression has been monitored using Affymetrix GeneChips Human Gene 1.0 ST array, using total RNA extracted from seven T2DM heart failure patients and five controls. The employing of the RMA approach was valuable in normalizing the data. The project aimed to discover markers that exhibited differential expression genes (DEGs) in these samples. The datasets were subjected to a comparative analysis using a *t*-test to identify genes exhibiting significant differential expression. A significance level of *p <* 0.05 was used. A heatmap diagram was constructed using the ggplot package (ggplot2_3.5.1), highlighting genes with a *p*-value < 0.001. Conversely, we used a log fold change (logFC) threshold of ±1.5 and ±0.15, respectively, to group the either overexpressed or downregulated genes, allowing us to select the significantly significant genes. In the next stage, we used the STRING 11.5 database to create a network of protein–protein interactions (PPIs) with hub nodes from each dataset study. We emphasized medium confidence, as defined by Szklarczyk et al. [41]. The hub genes were identified using CytoScape 3.6.0, with betweenness centrality, degree, and closeness centrality as criteria. This allowed for the visualization of the network characteristics [42]. The genetics network of hub nodes was designed based on network diameters, eigenvector centrality, and modularity class. The goal was to properly visualize and manipulate large graphs utilizing the Gephi software 9.2.0 platform [43]. Gephi is a powerful software application specifically created to visualize and analyze networks. The Enrich-KG and Reactome databases were used for gene set enrichment analysis to discover the crucial molecular signaling pathways and gene ontology processes associated with hub genes that had substantial differential expression in diabetic cardiomyopathy. Afterward, the genetic network, which includes these central genes, was created using Gephi software 9.2.0. We emphasized the shared genes between PBMC and heart tissue using a Venn visual design. To identify the extensive PPIs network of common hub genes associated with heart tissue and PBMC, we designed a genetic network in STRING version 11.5, using evidence and medium confidence (0.4). Utilizing Gephi 0.9.2 software, we displayed the network’s characteristics by configuring the network parameters, eigenvector centrality, and modularity class; moreover, we constructed the hub genes network based on betweenness centrality and modularity class. This work conducted a thorough analysis of many databases, including LncRNADisease [44], LncTard [45], and lncHUB2 [46]. Our study aimed to determine any connections between lncRNAs, hub genes, and diseases, particularly emphasizing their function as post-transcriptional regulatory agents. To build a hypothetical ceRNA network related to the development of colorectal cancer, we used the miRNet database [47] to find frequently expressed genes. Ultimately, we determined the most noteworthy lncRNA and microRNA inside the network by evaluating their degree and functional predictions. The predictions were obtained by examining the relationships between lncRNA and gene co-expression using the lncHUB service [46].

### 2.2. Molecular Docking Methodology

EGFR, CCL2, PTGS2 and SERPINE1 are categorized as master switching and druggable proteins in this genetic interaction network according to betweenness, centrality, and eigenvector structure evaluations. Hence, we have selected the EGFR, CCL2, PTGS2 and SERPINE1 proteins as primary targets for our pharmaceutical development and exploration endeavors aimed at alleviating the detrimental impact of diabetes on cardiac performance. An X-ray diffraction analysis was conducted to choose the most suitable three-dimensional structural model of EGFR, CCL2, PTGS2 and SERPINE1 using the protein data bank’s identifiers 4kro, 4zk9, 5F19 and 3LW2. Chimera v.1.8.1 software optimized the 3D crystal structure of EGFR, CCL2, PTGS2 and SERPINE1. The binding affinity score between bioactive compounds and EGFR, CCL2, PTGS2 and SERPINE1 was estimated by molecular docking modeling. An evaluation was conducted on the interaction of small compounds with the EGFR, CCL2, PTGS2 and SERPINE1 protein binding sites using PyRx software (PyRx-Python Prescription 0.8).

### 2.3. Animal Study

For this study, 7-week-old wild-type male C57BL/6 mice were obtained from the Royan Institute (Isfahan, Iran). Mice were allowed to acclimatize two weeks before the start of experiments. Mice weights were approximately 14 ± 2 g. Mice were divided into five groups (*N* = 6 per group): (1) a control group; (2) a T2DM group, induced by a 45% high-fat diet enriched with AGEs (45% HF-AGEs) over three months; (3) a T2DM group treated with metformin (T2DM+met, 300 mg/kg [48]) for eight weeks; (4) a T2DM group administered SBU (T2DM+SBU, 300 mg/mL) for eight weeks; and (5) a T2DM group receiving combination treatment of metformin and SBU for eight weeks (T2DM+met+SBU). Mice were maintained under controlled conditions at 23 °C with a 12 h light/dark cycle and supplemented with the respective diets and water ad libitum. Diets were provided using standard procedures as described elsewhere in the Royan Institute. The weights of the mice, calorie intake, and amount of drinking water were monitored weekly.

### 2.4. Mice Behavioral Tests

Several behavioral tests were implemented as follows:

#### 2.4.1. Elevated Plus Maze Assessment of Anxiety-Related Behavior

Anxiety-related behavior was evaluated using an elevated plus maze apparatus. The apparatus comprised two open arms (30 cm × 5 cm) and two enclosed arms of the same dimensions, connected by a central platform (5 cm × 5 cm). Mice were individually placed in the central area, and their behavior was recorded for one hour per day over three consecutive days. The primary measures were the number of entries and the total time spent in each arm. Increased time spent in and the number of entries into the closed arms were interpreted as indicators of heightened anxiety-like behavior, consistent with previous findings [49].

#### 2.4.2. Open-Field Test for Assessment of Locomotor Activity

Locomotor activity and rest time were assessed using an open-field test. The testing apparatus consisted of a 40 cm × 40 cm × 40 cm arena divided into 16 equal squares. Each mouse was individually placed in the center of the arena and allowed to explore freely for one hour per day over three consecutive days. Parameters recorded included total rest time (seconds), total distance traveled (arbitrary units), and total duration of movement (seconds).

### 2.5. Biochemical Tests

Blood samples were centrifuged at 4500 RPM for 15 min at 4 °C to separate serum. Serum concentrations of SERPINE1 (MyBioSource, MBS135529, San Diego, CA, USA), leptin (Crystal Chem, 90030, Elk Grove Village, IL, USA), adiponectin (Crystal Chem, 80569, Elk Grove Village, IL, USA), oncostatin (Abcam, ab263891, 152 Grove Street, Waltham, MA, USA), and insulin (Ultra-Sensitive Mouse Insulin, Crystal Chem, 90080) were also determined via ELISA according to manufacturer instructions. Following a 6 h fast, blood glucose was measured from the tail tip using an Alpha TRAK glucometer (Zoetis, Parsippany, NJ, USA). In the last months, intraperitoneal insulin tolerance testing (IPITT) and intraperitoneal glucose tolerance testing (IPGTT) were performed. For IPGTT, D-glucose (Sigma, Burlington, MA, USA; 0.75–1.5 g/kg body weight) was administered intraperitoneally. For IPITT, recombinant human insulin (0.5–1 U/kg body weight) was injected intraperitoneally. Blood glucose concentrations were measured from the tail vein at 0, 15, 30, 60, and 90 min post-injection using an Elite glucometer (Alpha TRAK, Zoetis, USA).

### 2.6. Quantitative Real-Time PCR (qRT-PCR)

Total RNA was extracted from left ventricle (LV) cardiac and PBMC using TRIzol reagent (Sigma, USA). cDNA synthesis was performed using 1 μg of total RNA and a cDNA synthesis kit (TaKaRa, Tokyo, Japan) according to the manufacturer’s instructions. qRT-PCR was conducted using SYBR Green (TaKaRa, Japan) on an Applied Biosystems StepOnePlus system (USA). Gene expression and lncRNAs were assessed using the 2^−ΔΔCT^ method, with values normalized to 18 s rRNA expression. All primers were synthesized by Metabion (Semmelweisstrasse, Planegg, Germany), and their sequences are provided in Table 1.

### 2.7. Protocol for Extracting Sea Buckthorn (Hippophae rhamnoides *L.*)

The sea buckthorn extract was obtained using established techniques, as previously reported [48]. In summary, the first wild elder was subjected to three boiling (6 h each) in ether at a temperature of 70 °C. Following air-drying in a ventilated cabinet, the material was incubated with 700 mL of ethanol at 80 °C for four cycles, each lasting two hours, to completely remove all crude flavones. The bulk of the crude flavonoids were then loaded onto a D101 macroporous resin column and separated using ethanol at a rate of 2 mL/min, starting from 300 mL/L and increasing to 500 mL/L [48,50]. Sea buckthorn extract was administered to mice by gavage at a dosage of 300 mg/kg body weight per day, five days a week, for a duration of eight weeks [48].

### 2.8. Metformin Administration

Metformin, a widely used medication for diabetes, decreases vascular calcification caused by hyperlipidemia due to its anti-iron properties. These findings indicate that the anti-ferroptotic properties of metformin may be beneficial in the management of diabetes. In addition to metformin, other natural compounds provide cellular protection against ferroptosis. Therefore, during a period of eight weeks, mice were orally administered a dose of 300 mg/kg of body-weight metformin on a daily basis (5 days per week) as a recognized and well-accepted treatment for diabetes [51].

### 2.9. Diagnostic Evaluation by Pathological Assessment

Following the sacrifice of the mice, the heart tissues were promptly preserved in a 10% buffered formalin solution and then embedded in paraffin. Furthermore, after the fixation process, the tissues were carefully cut into sections measuring 5 µm in thickness. Following deparaffinization and hydration, tissue slide staining was undertaken using Hematoxylin and Eosin (H&E) and Masson. Following that, were examined using light microscopy.

### 2.10. Statistical Analysis

Data were presented as the mean and standard deviation (SD). Statistical analysis used GraphPad Prism (Version 8; GraphPad Software). The Kolmogorov–Smirnov test was used for normalizing distribution, and variables were normally distributed. Data were analyzed by one-way analysis of variance (ANOVA) with Tukey’s post hoc test due to multiple comparisons. Differences at *p* < 0.05 were considered to be significant. Furthermore, the data were assessed by Pearson correlation to examine the link between oncostatin and non-coding RNAs. The results were deemed statistically significant when the *p*-values were less than 0.05.

## 3. Results

### 3.1. Applying System Biology Research and Computational Molecular Docking Insights

The examination of the GSE156993 dataset indicated that there were 1812 genes in the diabetic PBMC sample that had statistically significant variations in expression when compared to the healthy sample (*p*-value < 0.05). Genes that exhibit a statistically significant differential expression (*p*-value < 0.001) are shown in the heatmap diagram, which may be seen in Figure 1a. In the setting of diabetic heart disease, our research showed that there was a substantial downregulation of 519 genes and an overexpression of 388 genes in the PBMC. In addition, while implementing network parameters, we effectively identified 124 nodes that demonstrated the highest degree and betweenness centrality as hub genes related to the prevalence of diabetic heart disease. In addition, the genetic interaction network comprising 124 hub genes found that *EGFR*, *CCL2*, and *PTGS2* were the primary nodes with the highest betweenness centrality (Figure 1b). Through the enrichment of 124 hub genes, the molecular signaling pathways of hypertensive disease, congestive heart failure, heart failure, neoplasm and carcinogenesis, PI3K-Akt, endochondral ossification, extracellular organization, oncostatin M signaling, cardiac troponin T level, plasma plasminogen activation levels, and biological processes that are associated with these hub genes were brought to light (Figure 1c).

The analysis of the GSE26887 dataset revealed that 2844 genes in the diabetic cardiac sample had statistically significant expression differences compared to the healthy sample (*p*-value < 0.05). Genes with significant differential expression (*p*-value < 0.001) are highlighted in the heatmap diagram presented in Figure 2a. In the context of diabetic heart disease, our study revealed a significant downregulation of 176 genes and an upregulation of 351 genes in the left ventricular cardiac tissues. Furthermore, during the implementation of network parameters, we successfully identified 119 nodes exhibiting the greatest degree and betweenness centrality as hub genes associated with the frequency of diabetic heart disease. Furthermore, the genetic interaction network, consisting of 119 hub genes, identified *IL6*, *SPP1*, and *CD163* as the principal nodes exhibiting the greatest betweenness centrality (Figure 2b). The enrichment of 119 hub genes elucidated the molecular signaling pathways related to atherosclerosis, myocardial infarction, the PI3K-Akt signaling pathway, the HIF1 signaling pathway, the Toll-like receptor signaling pathway, IL-18 signaling and hypertrophic cardiomyopathy and biological processes associated with these hub genes (Figure 2c). Applying the Venn diagram tool identified that PBMC and LV cardiac samples provide cross-talk via the participation of six shared components: NRG1, NR4A2, SERPINE1, MYH11, PTH, and HMCN1 (Figure 2d). Thus, the construction of a genetics network of six common genes with medium confidence indicates that SERPINE1, displaying the highest level of connectivity and betweenness centrality, may be identified as a viable therapeutic target for diabetes-induced cardiac failure (Figure 2e). On the other hand, enrichment analysis of hub genes in the Reactome database indicated that these genes are associated with programmed cell death as the characteristic of diabetic cardiomyopathy.

Moreover, the differential transcript of lncRNAs as modifier post-translational molecules could play 363 a key role in pathogenesis and therapy. Therefore, based on diseases and target genes in the pathomechanism network of diabetic cardiomyopathy, we predicted a pool of lncRNAs such as GAS5, MALAT1, SNHG15, LOXL1-AS1, H19, MIAT, PRINS, TINCR, NEAT1, MEG3, TUG1, PVT1, GREM1, HOTAIR, and CASC5. A potential ceRNA network between common genes and predicted lncRNA was designed to highlight significant lncRNAs in cardiomyopathy pathogenesis (Figure 3).

We found that lncRNA MALAT1 and lncRNA NEAT1 target hub genes through the ceRNA network. Enrichment analysis of MALAT1 and NEAT1 in lncHUB showed that lncRNA MALAT1 is a regulatory factor in inflammation, oxidative phosphorylation, positive regulation of the NF-KB signaling pathway, regulation of transcription, positive regulation of the apoptotic signaling pathway in the absence of ligand, the NF-kB signaling pathway, the PI3k-Akt signaling pathway, and the IL-17 signaling pathway (Figure 3b–e). Moreover, NEAT1 is associated with increased circulation IL-18 level, chemical homeostasis within a tissue, regulation of vasculature development, cardiac muscular adhesion, bundle of His cell-PURKINJE myocyte adhesion involved in cell communication, external encapsulation structure organization, regulation of type 2 immune response, positive regulation of angiogenesis, regulation of the protein kinase B signaling pathway, abnormal inflammation, mitochondrial cytochrome c oxidase assembly, respiratory chain complex IV, the PI3K-Akt signaling pathway, transcriptional regulation, and oxidative phosphorylation (Figure 3f–i).

The results of our research suggest that EGFR, CCL2, PTGS2 and SERPINE1 proteins that play a significant role in the diabetic cardiomyopathy network may function as a druggable node to impact the repercussions of diabetes in cardiomyocytes. The molecular docking approach was able to ease the binding affinity calculation between the bioactive components of *Hippophae rhamnoides* L. extract and the macromolecule, as shown by our findings (Appendix A and Table 2, Table 3, Table 4 and Table 5).

### 3.2. Metabolic, Physiological, and Behavior Effects of SBU and Metformin in Diabetic Mice

A 45% HF-AGE diet led to significantly more weight gain than controls (Figure 4a). Additionally, mice consuming SBU and met exhibited reduced caloric intake relative to T2DM (Figure 4b). Mice fed the T2DM+met+SBU displayed decreased calorie intake relative to other groups (Figure 4b).

Open-field testing revealed decreased movement time, distance traveled, and increased rest time in the T2DM group, indicating reduced physical activity (Figure 4c,d). Moreover, the data indicated that movement time and distance traveled significantly increased in the T2DM+met and T2DM+SBU groups compared with the T2DM group (Figure 4c,d). In addition, the movement time and distance traveled were not changed between T2DM+met and T2DM+SBU groups (Figure 4c,d). Notably, the group received a combination of metformin and SBU for eight weeks (T2DM+met+SBU), significantly improving movement time and distance traveled (Figure 4c,d). The rest time was declined by the T2DM group treated with metformin (T2DM+met, 300 mg/kg); the T2DM group administered SBU (T2DM+SBU, 300 mg/mL) for eight weeks (Figure 4d). Elevated plus maze testing demonstrated increased anxiety-related behavior in the T2DM group, as evidenced by an increased number of entries into the closed arms (Figure 4e,f).

The glucose tolerance test (GTT) and insulin tolerance test (ITT) indicated impaired glucose response in the T2DM group compared to the control group (Figure 4g,h). Notably, the group received a combination of metformin and SBU for eight weeks (T2DM+met+SBU), significantly ameliorating the GTT and ITT (Figure 4g,h).

The serum levels of leptin, adiponectin, fasting blood glucose (FBS), and insulin were evaluated in this study (Figure 4i–l). The serum concentration of the leptin, insulin, and FBS was elevated compared to all other groups (Figure 4i–k). Furthermore, the adiponectin concentration declined in the T2DM group compared to the other group (Figure 4l). In this study, as expected, we found that the serum levels of leptin, insulin, adiponectin, and FBS concentration were ameliorated in diabetic mice treating metformin and SBU (Figure 4i–k). Interestingly, there was no change between the T2DM+SBU group and the T2DM+met group (Figure 4i–k). We found the synergistic effect on leptin, insulin, adiponectin, and FBS concentration (Figure 4i–l). Based on our data, leptin, insulin, and FBS concentrations declined, and adiponectin was increased in the T2DM+met+SBU group (Figure 4i–l).

In addition, we assessed the relative expression of the Gut4 mRNA muscle in different groups (Figure 4m). Recent evidence has indicated that the mRNA GLUT4 might be a marker for indicating T2D. We found that the expression level of the GLUT4 mRNA was reduced in the muscle of the T2DM group (Figure 4m). Moreover, metformin and SBU regulated the relative expression of the GLUT4 mRNA (Figure 4m). The T2DM group received a combination treatment of metformin and SBU for eight weeks (T2DM+met+SBU), significantly increasing the mRNA GLUT4 in the muscle of the T2DM+met+SBU group compared with the other group (Figure 4m).

In the subsequent phase of our study, we assessed heart failure induced by T2DM by examining cardiac tissue sections’ organization and structural integrity, alongside measuring atrial natriuretic peptide (ANP) concentrations.

### 3.3. The Pathological Features Were Evaluated in Diabetic Mice: SBU and Metformin Improved the Pathological Features in Diabetic Mice

Our findings revealed significant destruction of cardiac tissue in the T2DM group compared to the control group (Figure 5a,b), as evidenced by necrotic areas. As illustrated in Figure 5b, the cardiac tissue in the T2DM group displayed signs of failure, including cytoplasmic degeneration, vascular congestion, loss of striations, edema, and lymphocytic infiltration. Treatment with metformin and SBU notably reduced the extent of necrosis and the incidence of cell death characterized by nuclear loss (Figure 5c,d). Furthermore, the interaction between metformin and SBU demonstrated enhanced efficacy relative to other treatment groups (Figure 5e). Notably, the morphology of cardiac tissue in the T2DM+met+SBU group exhibited marked improvement compared to the T2DM group (Figure 5e). In addition, the T2DM+met+SBU group exhibited relative preservation of striations and signs of cytoplasmic regeneration (Figure 3e).

In the next step we evaluated the pathological feature via Masson staining. Based on Figure 5g, we indicated cardiac tissue failure, including cytoplasmic degeneration, congested vessels, loss of striation, congestion, edema, and lymphocytic infiltration. The data indicated necrosis areas in the T2DM group compared with the control group (Figure 5f,g). Our data indicated that metformin and SBU significantly decreased the necrosis areas and cell death with the loss of the nuclei (Figure 5h,i). In addition, the staining indicated that metformin and SBU had moderate organization and reduced fibrosis (Figure 5h,i). Moreover, we found that the metformin and SBU interaction (T2DM+met+SBU) was effective compared with other groups (Figure 5j). Notably, the morphology of the heart tissue treated with metformin and SBU improved compared with the T2DM group (Figure 5j). In the T2DM+met+SBU group, relative striation and cytoplasmic regeneration were found (Figure 5j).

### 3.4. The Pattern Expression of SEPRINE1/NRG1/MYH11/PTH/NR4A in the Heart of Diabetic Mice: SBU and Metformin Improved the SEPRINE1/NRG1/MYH11/PTH/NR4A Level

We assessed the relative expression of the and vital hub genes, which was selected via bioinformatic and data analysis in the heart tissue. Based on our data, we found that *SEPRINE1/NRG1/MYH11/PTH/NR4A* could be candidate the targeted genes to ameliorated the cardiac disfunction in the T2DM group. The data have indicated that the level mRNA of *SEPRINE1/NRG1/MYH11/PTH* significantly was increased in the T2DM group compared with the control (Figure 6a–e). Furthermore, the expression level of *SEPRINE1/NRG1/MYH11/PTH* was reduced by metformin (T2DM+met group) and SBU (T2DM+SBU group) compared with the T2DM group (Figure 6a–e). Moreover, the relative expression of the was unchanged between T2DM+met and T2DM+SBU groups (Figure 6a–e). Interestingly, we found that the expression level of *SEPRINE1/NRG1/MYH11/PTH* in heart tissue was regulated by the consuming combination of the met+SBU in the T2DM+met+SBU group in comparison with other groups (Figure 6a–e). Moreover, we evaluated the concentration of the SEPRINE1 in the serum of the diabetic mice treated with met and SBU. We found that the concentration of the SEPRINE1 was elevated in the T2DM group compared with the control group (Figure 6f). The SEPRINE1 concentration was reduced by met (T2DM+met group) and SBU (T2DM+SBU group). Notably, consuming met along with SBU (T2DM+met+SBU group) significantly decreased the SEPRINE1 concentration compared with the other groups (Figure 6f).

### 3.5. ANP Concentration Was Evaluated in Serum of Diabetic Mice: SBU and Metformin Improved the ANP Concentration in Diabetic Mice

The ANP concentration was enhanced in the T2DM group compared with the control group (Figure 6g). Hence, based on these data, we found that the heart of the diabetic mice was implicated. These data were in line with the pathological feature and expression level of *SEPRINE1/NRG1/MYH11/PTH/NR4A* in diabetic mice. Furthermore, the concentration of ANP was reduced in the T2DM+met and T2DM+SBU groups (Figure 6g). The results revealed that the ANP concentration was significantly decreased in the T2DM+met+SBU group compared to the other group (Figure 6g). Hence, the interaction between metformin and SBU has a synergetic effect on the pathological features and ANP concentration in diabetic mice.

### 3.6. Dysregulation of the NRF2/PGC1α/ATF1/Ascl2/NOX1/GPX4/NLRP3/CCK8/COX2/CCL2/PTGS2/EGFR Network in the PBMC of Diabetic Mice

Based on the artificial intelligence, we found that dysregulation of the *NRF2/PGC1α/ATF1/ASCL2/NOX1/GPX4/NLRP3/CCK8/COX2/CCL2/PTGS2/EGFR* hub genes might trigger the ferroptosis (Figure 6h–s). Hence, we assessed the expression level of these hub genes. Our data indicated that the *NRF2/PGC1α/GPX4/ATF1/Ascl2* involved the mitochondrial function, and oxidative stress declined in the heart of the diabetic mice (Figure 6h–l). Moreover, the mRNA level of the *NOX1/NLRP3/CCK8/COX2/CCL2/PTGS2/EGFR* associated with inflammatory responses, was elevated in the T2DM group compared with the control group (Figure 6m–s).

### 3.7. SBU and Metformin Mediated the NRF2/PGC1α/ATF1/ASCL2/NOX1/GPX4/NLRP3/CCK8/COX2/CCL2/PTGS2/EGFR Network in PBMC

Interestingly, the SBU and metformin significantly enhanced the *NRF2/PGC1α/GPX4/ATF1/Ascl2* expression level (Figure 6h–l) and reduced the NOX1/NLRP3/CCK8/COX2/Ccl2/PTGS2/EGFR (Figure 6m–s). It should be noted that the T2DM group received a combination treatment of metformin and SBU for eight weeks (T2DM+met+SBU group) amplified the *NRF2/PGC1α/ATF1/Ascl2* expression level (Figure 6h–l) and declined the NOX1/NLRP3/CCK8/COX2/Ccl2/PTGS2/EGFR (Figure 6m–s).

### 3.8. Oncostatin as a Cell Death Biomarker Was Increased in Serum of Diabetic Mice

Oncostatin (OSM) can induce oxidative stress in target cells, leading to increased levels of ROS. This oxidative environment can promote ferroptosis by enhancing lipid peroxidation (Figure 6t). Additionally, OSM signaling may affect the expression of key proteins involved in iron metabolism and antioxidant defense, further influencing the ferroptosis response. The results demonstrated that the concentration of the OSM was increased in the T2DM group compared with the control group (Figure 6t). Therefore, dysregulation of the *NRF2/PGC1α/ATF1/ASCL2/NOX1/GPX4/NLRP3/CCK8/COX2/CCL2/PTGS2/EGFR* hub genes might release the OSM and trigger the ferroptosis in the heart cells.

### 3.9. SBU and Metformin Regulated Oncostatin in Serum of Diabetic Mice

We found that the oncostatin concentration was decreased in the T2DM+met and T2DM+SBU group compared with the T2DM group (Figure 6t). Moreover, a combination treatment of metformin and SBU for eight weeks (T2DM+met+SBU group) decreased the oncostatin concentration in the serum (Figure 6t). These data were in line with the pathological features and ANP concentration and regulating the *NRF2/PGC1α/ATF1/ASCL2/NOX1/GPX4/NLRP3/CCK8/COX2/CCL2/PTGS2/EGFR* hub genes, which indicated the ferroptosis in the heart cells was declined.

### 3.10. The lncRNAs (NEAT1 and MALAT1) Are Implicated in the Diabetic Condition: SBU and Metformin Ameliorated the Expression of lncRNAs (NEAT1 and MALAT1) in the Heart Tissue

NEAT1 and MALAT1 were selected as vital regulators in heart failure induced by type 2 diabetes through bioinformatic analysis. In the next step, we measured the relative expression of lncRNAs (NEAT1 and MALAT1) (Figure 7a,b). We found that the expression level of the NEAT1 and MALAT1 was upregulated in the T2DM group compared with the control group (Figure 7a,b). Moreover, we found that the NEAT1 and MALAT1 expression was decreased by the T2DM group treated with metformin (T2DM+met, 300 mg/kg) for eight weeks; the T2DM group was administered SBU (T2DM+SBU, 300 mg/mL) for eight weeks (Figure 5a,b). Notably, we found that the expression levels of NEAT1 and MALAT1 were significantly upregulated in the T2DM+met+SBU group compared with the other groups (Figure 7a,b). Hence, based on our data, we have indicated that the expression level of the NEAT1 and MALAT1 was predominantly enhanced by combination treatment of metformin and SBU for eight weeks (Figure 7a,b).

### 3.11. The Correlation Between lncRNAs (NEAT1 and MALAT1) and Oncostatin

In this investigation, we examined the relationship between oncostatin concentration and the expression levels of NEAT1 and MALAT1 (Figure 6a,b). Our findings indicate a positive correlation between oncostatin concentration and the relative expression of both NEAT1 and MALAT1 (Figure 6a,b). Consequently, increased levels of oncostatin were associated with elevated expression levels of NEAT1 and MALAT1. These analyses reveal a robust positive correlation between oncostatin concentration and the expression levels of NEAT1 and MALAT1 (Figure 6a,b).

## 4. Discussion

This study aimed to explore the potential protective effects of *Hippophae rhamnoides* L. extract and metformin, both individually and in combination, against DCM in a mouse model of T2DM. Through bioinformatic analysis, behavioral tests, biochemical assessments, and histopathological evaluations, we uncovered important insights into the complex mechanisms underlying cardiac dysfunction in T2DM and how these interventions can ameliorate such complications. Our findings corroborate earlier studies demonstrating that a high-fat diet enriched with AGEs significantly impairs glucose metabolism, resulting in weight gain, increased anxiety, and reduced physical activity. These metabolic disturbances are known to contribute to the development of DCM, characterized by cardiomyocyte death, oxidative stress, and fibrosis. Treatment with SBU and metformin effectively mitigated these effects, with the combination therapy showing synergistic benefits over individual treatments in terms of glucose homeostasis, improved insulin sensitivity, and enhanced locomotor activity. This suggests that a combined therapeutic strategy targeting multiple pathways may offer superior benefits for managing diabetes-induced heart dysfunction. Histopathological analysis of cardiac tissue revealed significant necrotic changes in diabetic mice, with a marked increase in atrial natriuretic peptide (ANP), a biomarker of cardiac stress. Treatment with SBU and metformin attenuated these pathological changes, indicating their cardioprotective effects. Specifically, both treatments modulated the expression of hub genes involved in ferroptosis, a regulated form of cell death driven by lipid peroxidation and oxidative stress. We observed that diabetic mice had dysregulated expression of key ferroptosis-related genes (NRF2, PGC1α, GPX4, NOX1, and NLRP3), while SBU and metformin normalized these gene expressions, reducing oxidative damage and inflammation in PBMC. Moreover, we found that the relative expression level of EPRINE1/NRG1/MYH11/PTH/NR4A was dysregulated in the heart of diabetic mice. Our results also underscore the role of lncRNAs in the progression of DCM. Specifically, NEAT1 and MALAT1 were significantly upregulated in the T2DM group and were positively correlated with oncostatin, a marker of cell death and inflammation. Both SBU and metformin reduced the expression of these lncRNAs, suggesting a novel mechanism by which they mitigate diabetic cardiac damage through lncRNA modulation. Interestingly, the ferroptosis pathway, often underexplored in the context of DCM, emerged as a critical driver of cardiomyocyte loss in T2DM. The dysregulation of ferroptosis-related genes highlights the complex interplay between oxidative stress, lipid metabolism, and iron homeostasis in the diabetic heart. Our study demonstrates that SBU and metformin not only reduce ferroptosis but also enhance antioxidant defenses, restoring cellular homeostasis. The ability of metformin to attenuate iron overload and lipid peroxidation, along with the rich bioactive components of SBU, provides a robust therapeutic framework for tackling the iron-dependent cell death mechanisms that contribute to DCM.

Based on the bioinformatic analysis, the hub genes involved in the ferroptosis signaling pathway were explored. We discovered NRF2/PGC1α/ATF1/ASCL2/NOX1/GPX4/NLRP3/CCK8/COX2/CCL2/PTGS2/EGFR/oncostatin mapping, which could regulate the ferroptosis signaling pathway. The analysis of the NRF2, PGC1α, ATF1, ASCL2, NOX1, GPX4, NLRP3, CCK8, COX2, CCL2, PTGS2, EGFR and oncostatin signaling pathways revealed significant interactions that underscore their roles in ferroptosis [48]. NRF2 emerges as a critical regulator, promoting the expression of antioxidant genes that help mitigate oxidative stress [52]. GPX4 is particularly crucial as it detoxifies lipid peroxides, and its activity is essential for preventing ferroptosis; decreased GPX4 levels lead to heightened vulnerability to lipid peroxidation [53]. The collaboration between NOX1 and NLRP3 indicates a vital interplay where NOX1-generated reactive oxygen species can activate NLRP3 inflammasome, thereby enhancing inflammatory responses that may potentiate ferroptosis cell death under certain conditions [54].

Additionally, PGC1α regulates mitochondrial function and energy metabolism, which are critical for maintaining the cellular antioxidant capacity [55]. The pathways involving ATF1 and ASCL2 provide further regulatory layers, influencing the cellular response to stress and adaptation, potentially impacting ferroptosis sensitivity [56,57]. Moreover, COX2 and oncostatin are implicated in inflammatory signaling that may modulate ferroptosis pathways, indicating that a balanced inflammatory environment is essential for cell survival. Collectively, these signaling components illustrate a complex network that governs cellular responses to oxidative stress and highlights the intricate regulation of ferroptosis in connection with metabolic and inflammatory states. The CCL2, PTGS2, and EGFR signaling pathways play interconnected roles in regulating ferroptosis, a form of regulated cell death characterized by the accumulation of lipid peroxides. CCL2, a chemokine that recruits immune cells to sites of inflammation, contributes to an inflammatory microenvironment that enhances oxidative stress, thereby promoting ferroptosis through increased lipid peroxide levels. PTGS2, or COX-2, is an enzyme that synthesizes prostaglandins from arachidonic acid and facilitates lipid peroxidation, further driving the ferroptosis process. Meanwhile, EGFR, a receptor that regulates cell survival and proliferation, can counteract ferroptosis by promoting cell survival pathways [58]; however, it also influences oxidative stress responses and modulates the expression of CCL2 and PTGS2, linking it to ferroptosis [59]. Together, these pathways illustrate the complex interplay between inflammation, oxidative stress, and cell death mechanisms, highlighting potential therapeutic targets for diseases where ferroptosis is critical.

The combination of SBU and metformin also showed promising effects on behavioral parameters, including reduced anxiety-like behavior and improved physical performance, further linking metabolic regulation with cardiac and mental health in diabetic models. The improved metabolic and cardiac outcomes suggest that the dual approach of combining natural compounds like SBU with traditional anti-diabetic drugs such as metformin holds great promise in DCM management. In summary, our findings suggest that SBU and metformin, particularly in combination, offer significant therapeutic potential for alleviating the metabolic, inflammatory, and oxidative stress-related changes observed in T2DM-induced cardiac dysfunction. Further research is warranted to explore the exact molecular mechanisms driving these effects and to assess their clinical applicability in managing diabetic cardiomyopathy.

DCM is now widely acknowledged as a major consequence of T2DM, which substantially increases the likelihood of heart failure. The multifactorial etiology of DCM has been extensively studied over the last decade, revealing that metabolic dysregulation, oxidative stress, inflammation, and altered lipid metabolism are crucial in its development. Emerging research has underscored the significance of ferroptosis, oxidative stress, and lncRNAs in the development of DCM, therefore identifying new areas for therapeutic intervention. In recent years, ferroptosis, a controlled phenomenon of cell death induced by iron-dependent lipid peroxidation, has attracted interest due to its involvement in certain cardiovascular disorders. Ferroptosis has been shown by Ma et al. (2020) to play a substantial role in cardiac ischemia/reperfusion (I/R) damage and myocardial infarction [60]. In this context, the excessive buildup of iron results in increased oxidative stress and lipid peroxidation in cardiomyocytes. Furthermore, Zhao et al. (2023) showed that suppressing ferroptosis in models of diabetic cardiomyopathy decreases the generation of ROS, thereby enhancing cardiac function [61]. Consistent with previous research, we found that both SBU and metformin reduced ferroptosis in the diabetic heart by modulating several genes associated to ferroptosis, including NRF2, GPX4, and NOX1.

The evolution of DCM is mostly influenced by oxidative stress. A study by Zhao et al. (2023) shown that oxidative stress resulting from high blood sugar and low lipid levels worsens DCM by reducing mitochondrial activity and initiating apoptosis in cardiomyocytes [62]. Their result highlights the capacity of antioxidant treatments, like activators of the NRF2 pathway, to reduce cardiac oxidative tissue damage. Conforming to these results, our investigation showed that both SBU and metformin increased the expression of NRF2, thereby improving the cellular antioxidant response and inhibiting oxidative damage in the diabetic heart. This observation implies that the antioxidant characteristics of SBU and the anti-iron therapeutic benefits of metformin could work together to effectively counteract ferroptosis and oxidative stress.

Recent investigations have examined the function of lncRNAs in heart disease. NEAT1 and MALAT1 have been associated with the development of cardiac fibrosis, inflammation, and tissue hypertrophy. A study conducted by Ginckels et al. (2022) revealed that NEAT1 stimulates heart inflammation in diabetes mice, while MALAT1 regulates oxidative stress responses [62]. Furthermore, Cai et al. (2022) discovered that the ceRNA network, which includes long non-coding RNAs (lncRNAs) and their interactions with microRNAs, plays a crucial role in controlling cardiac fibrosis and ferroptosis [63]. Our work confirms these findings by demonstrating the increased expression of NEAT1 and MALAT1 in diabetic mice and their association with elevated levels of oncostatin, a biological marker of inflammation. The downregulation of these long non-coding RNAs by both SBU and metformin successfully mitigated heart injury. Therefore, modulating lncRNAs may be a crucial therapeutic approach in controlling cardiomyopathy caused by T2DM.

Early research findings have shown that both natural substances and metformin have therapeutic promise in the management of diabetes-related problems, including DCM [64]. As the first-line treatment for T2DM, metformin is well recognized for its ability to enhance insulin sensitivity, decrease hepatic glucose production, and provide cardiovascular protection. The anti-inflammatory, anti-fibrotic, and antioxidant effects of metformin have been emphasized in many studies, demonstrating their significant advantages in mitigating heart dysfunction in individuals with DCM [64].

Specifically, Ren Shang et al. (2023) examined the effect of metformin in decreasing oxidative stress and enhancing mitochondrial activity, which is crucial in halting the advancement of DCM [65]. Furthermore, metformin has been shown to regulate lipid metabolism and decrease the buildup of AGEs, a key factor in the development of heart damage caused by diabetes [66]. The results of our study align with previous research, as we demonstrated a significant decrease in indicators of oxidative stress and better regulation of glucose levels after administering metformin to diabetic mice.

Bioactive chemicals, particularly those with antioxidant characteristics, have attracted attention for their ability to control DCM. A study by Chen et al. (2020) showed that plant extracts abundant in flavonoids may regulate lipid metabolism and decrease cardiomyocyte death in diabetes mice [16,67]. The cardioprotective and anti-inflammatory benefits of sea buckthorn (*Hippophae rhamnoides* L.), which is abundant in bioactive substances such as flavonoids, carotenoids, and polyphenols, have been investigated. Research conducted by Xu et al. (2020) showed that the extract of sea buckthorn reduced oxidative stress and enhanced heart function in diabetic rats [68]. The results of our study are consistent with these observations, as the administration of SBU to diabetic mice led to significant enhancements in heart function, a reduction in inflammation, and modulation of crucial genes associated with ferroptosis. Employing natural bioactive substances such as SBU for the management of diabetes and its associated problems has become more popular. Kopčeková et al. (2023) established the antioxidant, anti-inflammatory, and cardioprotective properties of sea buckthorn extract in a rat model of diabetes. The results showed significant enhancements in glucose metabolism and lipid profile [69].

The present study provides more evidence for the combined benefits of natural chemicals and metformin in the management of DCM. Although both therapies alone increased glucose tolerance, decreased inflammation, and improved heart function, the combination of metformin and SBU had a more significant therapeutic impact. This is consistent with the results of Zhao et al. (2019), who demonstrated that the combination of natural antioxidants with conventional diabetes drugs may improve treatment results by addressing distinct pathogenic pathways concurrently [70].

Interestingly, bioinformatic analysis and subsequent experimental validation identified a network of hub genes (NRF2/PGC1A/ATF1/ASCL2/NOX1/GPX4/NLRP3/CCK8/COX2/CCL2/PTGS2/EGFR) involved in regulating ferroptosis. Our findings suggest that T2DM disrupts this intricate network, leading to increased oxidative stress, inflammation, and, ultimately, cardiomyocyte ferroptosis. SBU and metformin effectively modulated the expression of these hub genes, restoring the balance between pro- and anti-ferroptotic signaling pathways. This modulation likely contributes to the observed cardioprotective effects of these treatments. Further investigation revealed the involvement of lncRNAs, specifically NEAT1 and MALAT1, in T2DM-induced cardiac dysfunction. These lncRNAs were significantly upregulated in the T2DM group, and their expression levels were positively correlated with serum oncostatin concentrations, a marker of cell death and inflammation. Treatment with SBU and metformin effectively downregulated NEAT1 and MALAT1 expression, suggesting a potential mechanism by which these treatments mitigate cardiac damage.

ATF1 (Activating Transcription Factor 1) is a protein that plays a role in cell survival during early development. Studies have shown that a reduction in ATF1 leads to increased cell death. Oncostatin M (OSM) is a cytokine belonging to the interleukin-6 (IL-6) family, which is integral to numerous physiological and pathological processes, including inflammation, cell proliferation, and differentiation. Recent investigations have elucidated the role of OSM in regulating cell death mechanisms, particularly ferroptosis—a distinct form of regulated cell death characterized by the accumulation of lipid peroxides to toxic levels. Studies have demonstrated that OSM can induce oxidative stress in target cells, resulting in elevated levels of ROS. This oxidative milieu can facilitate ferroptosis by promoting lipid peroxidation. Furthermore, OSM signaling may alter the expression of critical proteins involved in iron metabolism and antioxidant defense, impacting the ferroptotic response.

The interplay between OSM and ferroptosis has gained attention in various disease contexts, including cancer and neurodegenerative disorders. In certain malignancies, OSM may contribute to tumor progression by fostering ferroptosis resistance, enabling cancer cells to survive under otherwise lethal conditions. Conversely, in neurodegenerative diseases, OSM-induced ferroptosis may be implicated in neuronal cell death, thereby contributing to disease pathology. The relationship between oncostatin M and ferroptosis highlights the complexity of cell death regulation and suggests potential therapeutic strategies. Targeting OSM signaling pathways may provide novel approaches for modulating ferroptosis in diverse diseases, representing a promising area for future research.

### Limitations and Future Perspectives

The present work applied a murine model of T2DM to examine DCM, a paradigm that may not completely reproduce the intricacy of human DCM. While animal models provide useful insights, the substantial physiological disparities between mice and humans might restrict the applicability of the results to clinical environments. Notably, while mice have comparable metabolic and cardiac responses to diabetes, variables such as longevity, cardiac physiology, and metabolic rates vary from those seen in humans. The duration of the SBU and metformin supplementation intervention was eight weeks, which may not accurately represent the extended-term effects of these therapies. DCM is a condition that advances over time, and short-term studies may not completely accurately represent the chronic character of heart failure caused by diabetes. Thus, the long-term safety and effectiveness of SBU and metformin, especially their effect on long-term outcomes such as survival and quality of life, have not been investigated. Despite the identification of important genes and pathways related to ferroptosis and lncRNA regulation in this work, the precise molecular mechanisms by which SBU and metformin exercise their protective effects remain incompletely known. Further investigation is required to examine the molecular and cellular interactions of these therapies, including their impact on other types of programmed cell death such as apoptosis and necroptosis. The lack of clinical trials or human investigations on the combination of SBU and metformin hinders our capacity to provide definitive recommendations about their suitability for human use. Physiological kinetics, bioavailability, and possible adverse effects of SBU, when combined with metformin, must be assessed in clinical settings before to making therapeutic recommendations. In order to confirm the efficacy and safety of combining SBU with metformin in the treatment of DCM, future research should prioritize the execution of long-term clinical studies in diabetic patients. In order to determine the therapeutic significance of this combination in people, these studies should examine cardiovascular outcomes such as heart failure progression, myocardial fibrosis, and overall survival rates.

Although the present work primarily examined sea buckthorn, it is recommended to investigate alternative natural substances that possess established anti-inflammatory, antioxidant, or anti-ferroptotic characteristics in conjunction with metformin. Subsequent investigations may ascertain other bioactive substances obtained from plants and assess their possible synergistic impacts in addressing the heart damage caused by diabetes. Additional comprehensive mechanistic investigations are necessary to completely comprehend the mechanisms by which SBU and metformin suppress ferroptosis and regulate the production of lncRNAs in diabetic hearts. Investigation of gene editing technologies like CRISPR-Cas9 might facilitate the identification of crucial gene targets, pathways, and regulatory mechanisms responsible for these protective effects. Furthermore, investigations using genetic knockout models might clarify the precise functions of ferroptosis-related genes (GPX4 and NRF2) and long non-coding RNAs (MALAT1 and NEAT1) in the development of DCM. The use of multi-omics methodologies (such as proteomics, transcriptomics, and metabolomics) may provide a more holistic perspective on the molecular alterations taking place in DCM after treatment with SBU and metformin. Integrated, these data have the potential to provide new biomarkers and therapeutic targets for DCM, therefore facilitating the development of more individualized treatment approaches. The findings of this research indicate that the combination of SBU with metformin may improve the therapeutic results in comparison to single administration of either therapy. Further investigation should include the examination of other combination therapy, such as the incorporation of SBU with contemporary diabetic drugs such SGLT2 inhibitors or GLP-1 receptor agonists. Such combinations might provide more comprehensive strategies for regulating both glycemic control and cardiovascular outcomes in type 2 diabetes mellitus. Potential future research might include the advancement of innovative drug delivery methods for SBU and metformin, like formulations based on nanoparticles or systems with controlled-release mechanisms. The use of these technologies has the potential to augment the bioavailability and therapeutic effectiveness of the medicaments, therefore yielding improved patient outcomes with less adverse effects. To summarize, our study offers significant insights into the therapeutic feasibility of sea buckthorn and metformin in treating diabetic cardiomyopathy. However, additional research is necessary to validate these results, particularly in long-term human trials, and to completely understand the molecular mechanisms responsible for their advantageous effects. The current study offered useful insights into behavioral, pathological, physiological, and metabolic markers but mostly concentrated on these aspects, neglecting comprehensive functional indices of heart performance, such as measured the blood pressure, ejection fraction (EF) and fractional shortening (FS). These tests are essential for assessing the extent of heart dysfunction potentially associated with diabetes-induced cardiomyopathy. It should be noted that the reactive oxygen species and cholesterol and triglyceride levels should be measured in future studies.

## 5. Conclusions

Our study provides compelling evidence for the therapeutic potential of SBU and metformin, particularly in combination, against T2DM-induced cardiac dysfunction. These beneficial effects are mediated by the modulation of multiple pathways, including regulating ferroptosis, key signaling molecules, and lncRNA expression. Further research is warranted to elucidate the precise molecular mechanisms underlying these protective effects and explore SBU and metformin’s clinical applicability as potential therapeutic strategies for diabetic cardiomyopathy. These results emphasize the possibility of integrating natural chemicals with traditional treatments to address many components of heart damage caused by diabetes. More clinical trials are recommended to investigate the therapeutic potential of this combination in human patients with diabetic cardiomyopathy fully.

## Figures and Tables

**Figure 1 biology-14-00361-f001:**
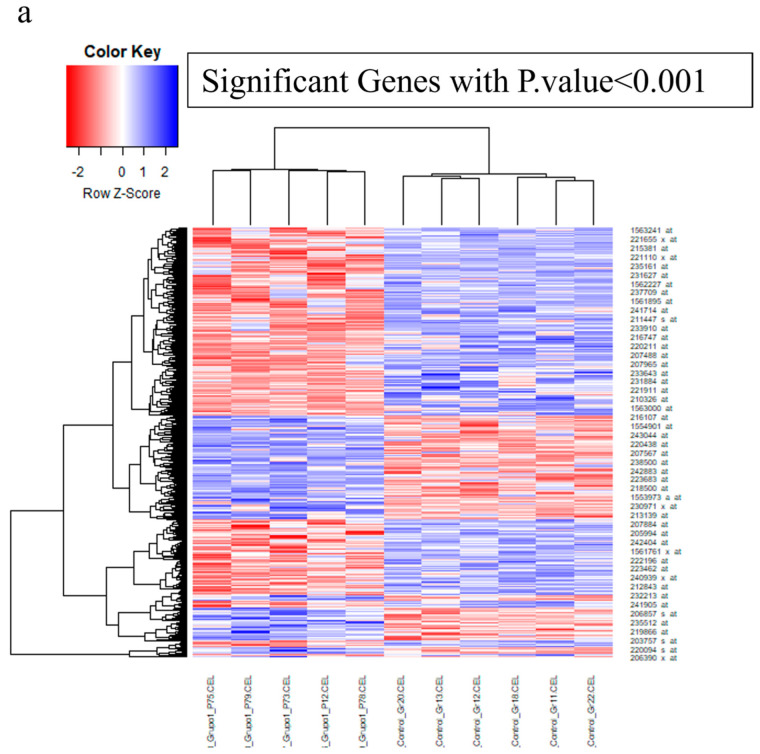
The present analysis applied the network visualization approaches to examine gene expression data, designing a model that depicts the progression of DCM. (**a**) Heatmap showing the differential expression of 1812 genes in the diabetic PBMC sample compared to the healthy sample. Genes with a *p*-value < 0.001 are displayed, with 519 genes downregulated and 388 genes overexpressed. (**b**) The genetic interaction network of 124 hub genes identified in diabetic heart disease, with *EGFR*, *CCL2*, and *PTGS2* highlighted as the primary nodes based on the highest betweenness centrality. (**c**) Gene set enrichment analysis of 124 hub genes in diabetic cardiomyopathy (DCM); The figure highlights the enriched molecular signaling pathways and biological processes associated with the 124 hub genes identified in DCM. Key pathways include those involved in hypertensive disease, congestive heart failure, neoplasm and carcinogenesis, PI3K-Akt signaling, endochondral ossification, extracellular matrix organization, oncostatin M signaling, cardiac troponin T levels, and plasma plasminogen activation. These pathways are integral to understanding the molecular mechanisms underlying DCM and offer potential therapeutic targets.

**Figure 2 biology-14-00361-f002:**
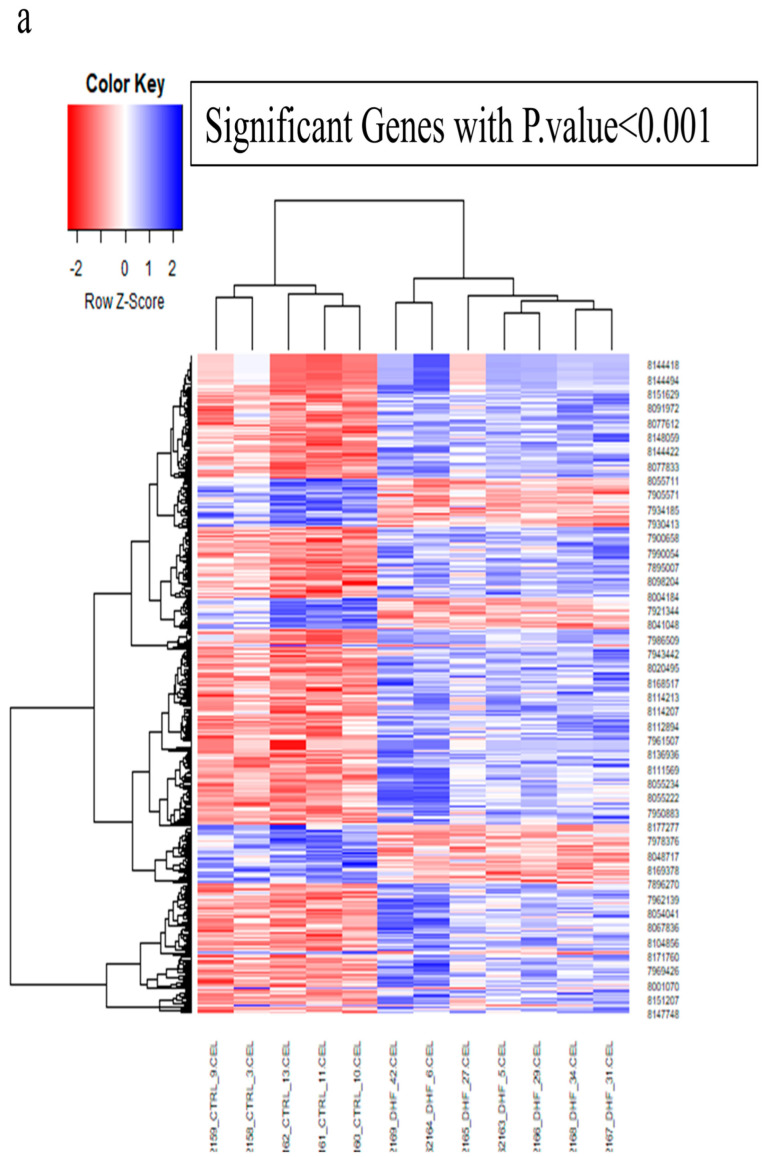
The research employing network visualization tools to analyze gene expression data, creating a model that highlights the advancement of DCM. (**a**) A heatmap illustrating the differential expression of 2844 genes in the diabetes PBMC sample relative to the healthy sample. Genes with a *p*-value less than 0.001 are shown, including 176 downregulated genes and 351 overexpressed genes. (**b**) The genetic interaction network comprising 119 hub genes found in diabetic heart disease, with IL6, SPP1, and CD163 emphasized as the principal nodes due to their greatest betweenness centrality. (**c**) Gene set enrichment analysis of 119 hub genes in diabetic cardiomyopathy (DCM); the figure illustrates the enriched molecular signaling pathways and biological processes linked to the 119 hub genes found in DCM. Principal pathways including those implicated in atherosclerosis, myocardial infarction, the PI3K-Akt signaling system, the HIF1 signaling route, the Toll-like receptor signaling pathway, IL-18 signaling, and hypertrophic cardiomyopathy. These pathways are essential for understanding the molecular causes of DCM and provide possible treatment targets. (**d**) The Venn diagram analysis revealed that PBMC and LV cardiac samples engage in cross-talk via six common factors: *NRG1, NR4A2, SERPINE1, MYH11, PTH, and HMCN1*. (**e**) The establishment of a genetics network of six prevalent genes with moderate confidence suggests that SERPINE1, exhibiting the greatest connectivity and betweenness centrality, should be recognized as a potential therapeutic target for diabetes-related heart failure. (**f**) An enrichment assessment of hub genes in the Reactome database revealed their association with programmed cell death, an important feature of diabetic cardiomyopathy.

**Figure 3 biology-14-00361-f003:**
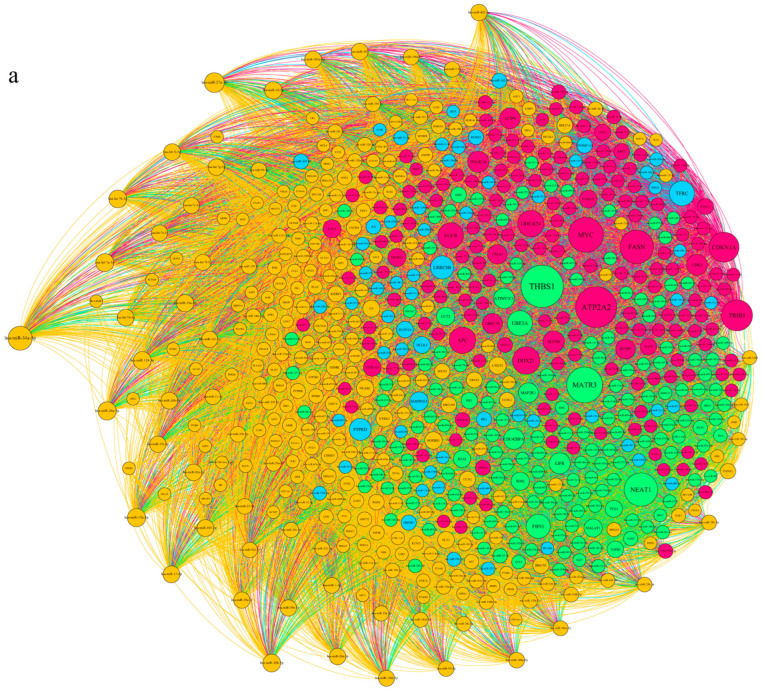
Construction of a ceRNA network in cardiomyopathy pathogenesis (**a**) A prospective ceRNA network including common genes and anticipated lncRNAs and microRNAs was established to emphasize crucial lncRNAs in the etiology of cardiomyopathy. (**b**–**e**) Enrichment analysis of lncRNA MALAT1 in lncHUB: The figure illustrates the molecular pathways regulated by lncRNA MALAT1, including inflammation, oxidative phosphorylation, the NF-kB signaling pathway, regulation of transcription, apoptotic signaling, the PI3K-Akt pathway, and the IL-17 signaling pathway. These pathways highlight MALAT1’s role in regulating key inflammatory and metabolic processes in diabetic cardiomyopathy. (**f**–**i**) Enrichment analysis of lncRNA NEAT1 in lncHUB: The figure shows the biological processes associated with lncRNA NEAT1, such as IL-18 circulation, vasculature development, cardiac muscular adhesion, regulation of type 2 immune response, angiogenesis, PI3K-Akt signaling, mitochondrial function, and oxidative phosphorylation. NEAT1 plays a significant role in inflammation and metabolic regulation, contributing to the pathogenesis of diabetic cardiomyopathy.

**Figure 4 biology-14-00361-f004:**
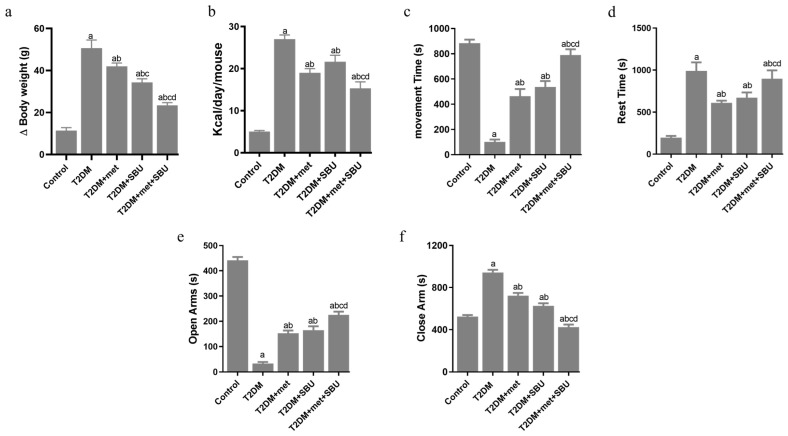
Phenotypic verification of induced T2D manifested by physical inactivity amplified depression and anxiety in mice. (**a**) ϴ gain (g) (mean ± SD; *n* = 6). (**b**) The measured calorie intake of each group of mice (mean ± SD; *n* = 6). The mice’s weight gain and calorie intake were measured every week (mean ± SD; *n* = 6)). (**c**) Movement time (s) and (**d**) rest time (s) in the open-field test. (**e**) Open arms (s) and (**f**) closed arms (s) in the plus elevated test (mean ± SD; *n* = 6). (**g**,**h**) Glucose test tolerance and insulin test tolerance were examined in all experimental groups. D-glucose (Sigma, 0.75–1.5 g per kg of body weight) and recombinant human insulin (0.5–1 U per kg of body weight) were injected intraperitoneally. Glucose concentration was measured by sampling a drop of blood from the tail with an Elite glucometer at 0, 15, 30, 60, and 90 min after injection (mean ± SD; *n* = 6). (**i**–**l**) serum leptin, insulin, FBS level, adiponectin (mean ± SD; *n* = 6). (**m**) (C) qRT-PCR for relative *Glut4/18s rRNA* expression was performed in skeletal muscle (gastrocnemius muscle). a Determines a significant difference with the control group at *p* < 0.05, b Determines significant difference with the T2DM group at *p* < 0.05, c Determines significant difference with the T2DM+met group at *p* < 0.05, and d Determines significant difference with T2DM+SBU group at *p* < 0.05. Data were analyzed through one-way analysis of variance (ANOVA) and Tukey’s post hoc test.

**Figure 5 biology-14-00361-f005:**
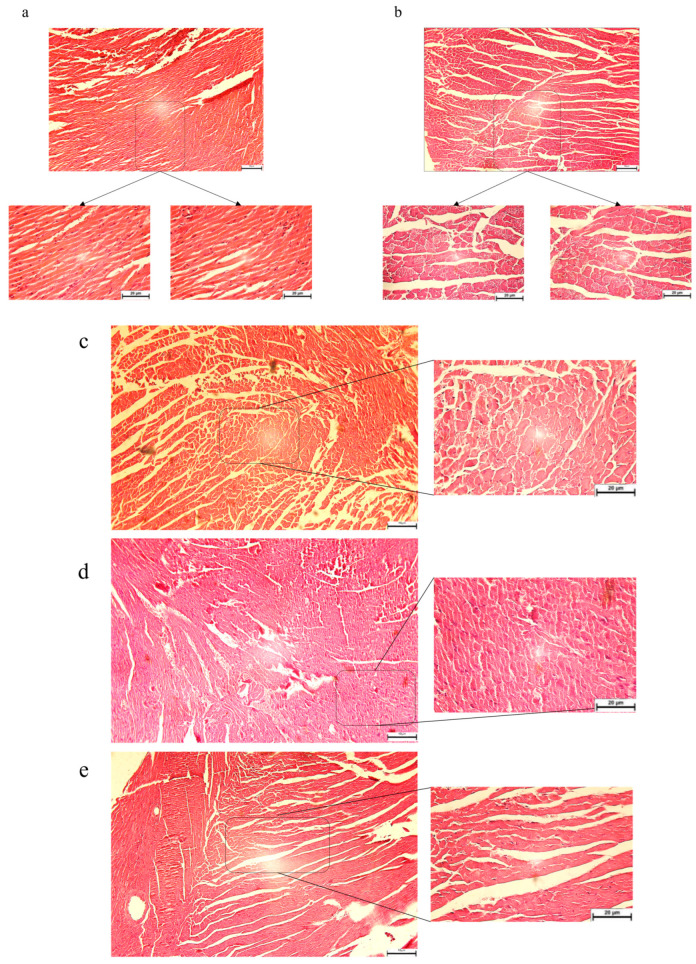
(**a**–**e**) Evaluation of the organization and structural integrity via pathological assay of heart tissue in different groups vi Hematoxylin and Eosin (H&E). (**f**–**j**) The histologic analysis of heart tissue in different groups via Masson staining. Scale bars: 10 and 20 µm.

**Figure 6 biology-14-00361-f006:**
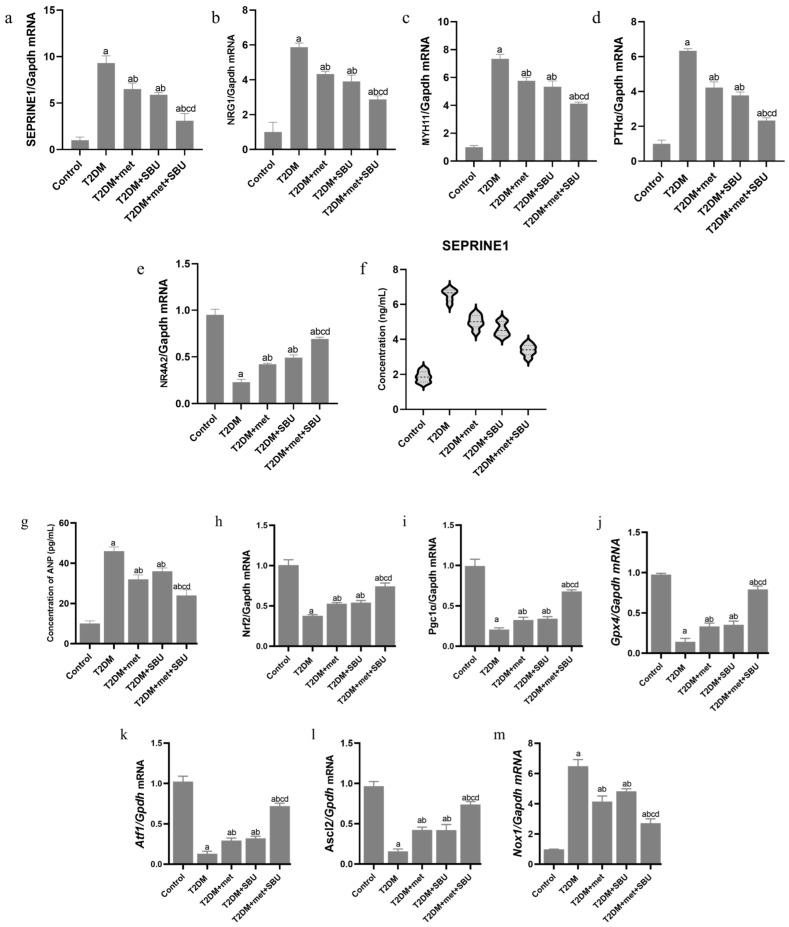
(**a**–**e**) mRNA level of *SEPRINE1/NRG1/MYH11/PTH/*NR4A in the heart tissue. (**f**) The concentration of the SEPRINE1 was measured by ELISA in the serum. (**g**) The ANP concentration was measured by ELISA in the serum. (**h**–**s**) The relative expression of the *NRF2/PGC1α/ATF1/Ascl2/NOX1/GPX4/NLRP3/CCK8/COX2/CCL2/PTGS2/EGFR* in the PBMC. (**t**) Oncostatin was measured by ELISA in the serum. a Determines a significant difference with the control group at *p* < 0.05, b Determines significant difference with the T2DM group at *p* < 0.05, c Determines significant difference with the T2DM+met group at *p* < 0.05, and d Determines significant difference with T2DM+SBU group at *p* < 0.05. Data were analyzed through one-way analysis of variance (ANOVA) and Tukey’s post hoc test.

**Figure 7 biology-14-00361-f007:**
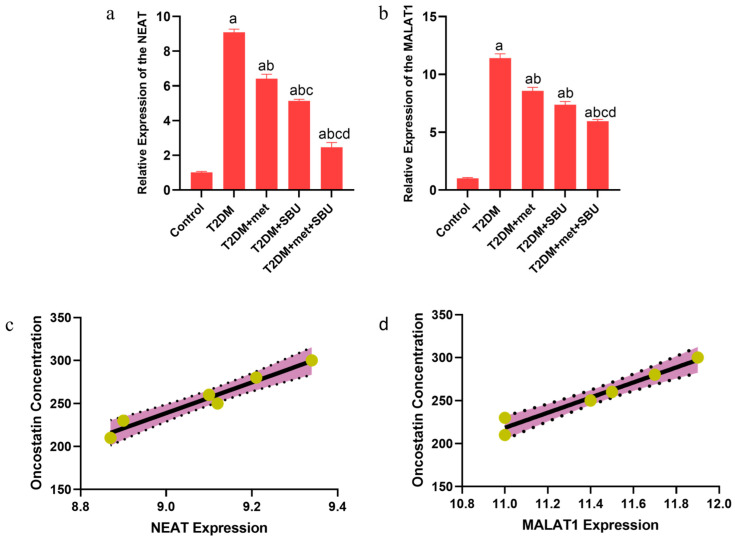
(**a**,**b**), Diagnostic index of NEAT1 and MALAT1 as biomarkers: NEAT1 and MALAT1 candidates as biomarkers in heart failure induced by type 2 diabetes. Moreover, SBU and met regulated the ceRNAs expression level. a. Expression level of the NEAT1. b. Expression level of the MALAT1. a Determines a significant difference with the control group at *p* < 0.05, b Determines significant difference with the T2DM group at *p* < 0.05, c Determines significant difference with the T2DM+met group at *p* < 0.05, and d Determines significant difference with T2DM+SBU group at *p* < 0.05. (**c**,**d**), Correlation between lncRNAs (NEAT and MALAT1) and oncostatin concentration.

**Table 1 biology-14-00361-t001:** The sequence of primers used in this study.

Gene	Sequence of Primers	Annealing Temperature
*NRF2*	Forward: 5′-CAGCATAGAGCAGGACATGGAG-3′	60 °C
	Reverse: 5′-GAACAGCGGTAGTATCAGCCAG-3′	
*PGC1α*	Forward: 5′-GAATCAAGCCACTACAGACACCG-3′	60 °C
	Reverse: 5′-CATCCCTCTTGAGCCTTTCGTG-3′	
*ATF1*	Forward: 5′-ATCAGACGAGCAGCGGACAGTA-3′	60 °C
	Reverse: 5′-GTCAGAGGTCTGTGCATACTGG-3′	
*ASCL2*	Forward: 5′-TTTCCTGTGCCGCACCAGAACT-3′	60 °C
	Reverse: 5′-CAGCGACTCCAGACGAGGTGG-3′	
*NOX1*	Forward: 5′-CTCCAGCCTATCTCATCCTGAG-3′	60 °C
	Reverse: 5′-AGTGGCAATCACTCCAGTAAGGC-3′	
*GPX4*	Forward: 5′-CCTCTGCTGCAAGAGCCTCCC-3′	60 °C
	Reverse: 5′-CTTATCCAGGCAGACCATGTGC-3′	
*NLRP3*	Forward: 5′-TCACAACTCGCCCAAGGAGGAA-3′	60 °C
	Reverse: 5′-AAGAGACCACGGCAGAAGCTAG-3′	
*CCK8*	Forward: 5′-TAGCGCGATACATCCAGCAGGT-3′	60 °C
	Reverse: 5′-GGTATTCGTAGTCCTCGGCACT-3′	
*CCL2*	Forward: 5′- GCTACAAGAGGATCACCAGCAG-3′	60 °C
	Reverse: 5′- GTCTGGACCCATTCCTTCTTGG-3′	
*PTGS2*	Forward: 5′-GCGACATACTCAAGCAGGAGCA-3′	60 °C
	Reverse: 5′-AGTGGTAACCGCTCAGGTGTTG-3′	
*EGFR*	Forward: 5′-GGACTGTGTCTCCTGCCAGAAT-3′	60 °C
	Reverse: 5′- GGCAGACATTCTGGATGGCACT-3′	
*GLUT4*	Forward: 5′-GGTGTGGTCAATACGGTCTTCAC-3′	60 °C
	Reverse: 5′-AGCAGAGCCACGGTCATCAAGA-3′	
*SERPINE1*	Forward: 5′-CCTCTTCCACAAGTCTGATGGC-3′	60 °C
	Reverse: 5′-GCAGTTCCACAACGTCATACTCG-3′	
*NRG1*	Forward: 5′-GCTCATCACTCCACGACTGTCA-3′	60 °C
	Reverse: 5′-TGCCTGCTGTTCTCTACCGATG-3′	
*MYH11*	Forward: 5′-GCAACTACAGGCTGAGAGGAAG-3′	60 °C
	Reverse: 5′-TCAGCCGTGACCTTCTCTAGCT-3′	
*PTH-* *α*	Forward: 5′-CAAACGGATGGGAAACCCGTGA-3′	60 °C
	Reverse: 5′-TGGCAGCCATTTGGACTCCAAG-3′	
*NR4A2*	Forward: 5′-CCGCCGAAATCGTTGTCAGTAC-3′	60 °C
	Reverse: 5′-TTCGGCTTCGAGGGTAAACGAC-3′	
*NEAT1*	Forward: 5′-GCTCTGGGACCTTCGTGACTCT-3′	60 °C
	Reverse: 5′-CTGCCTTGGCTTGGAAATGTAA-3′	
*MALAT1*	Forward: 5′-GGCAGAATGCCTTTGAAGAG-3′	60 °C
	Reverse: 5′-GGTCAGCTGCCAATGCTAGT-3′	
*18S rRNA*	Forward: 5′-CGGACACGGACAGGATTG-3′	59 °C
	Reverse: 5′-TCGCTCCACCAACTAAGAAC-3′	
*GAPDH*	Forward: 5′-TGCCGCCTGGAGAAACC-3′	60 °C
	Reverse: 5′-TGAAGTCGCAGGAGACAACC-3′	

**Table 2 biology-14-00361-t002:** Molecular docking analysis of bioactive components from *Hippophae rhamnoides* L. with PTGS2 protein: The tables present the binding affinity calculations between the bioactive compounds from *Hippophae rhamnoides* L. (sea buckthorn) and the target protein PTGS2, which are identified as key druggable nodes in the diabetic cardiomyopathy network. The docking results show the potential interactions and binding strengths, indicating the therapeutic potential of sea buckthorn components in modulating these proteins and mitigating the effects of diabetes on cardiomyocytes.

Macromolecule	Phytochemical Compound PubChem ID	RMSD Upper Bound	RMSD Lower Bound	Molecular Docking: Binding Affinity (kcal/mol)
PTGS	PTGS2PDB_157822370_uff_E = 101.29	0	0	−5.5
	PTGS2PDB_54670067_uff_E = 200.65	0	0	−5.8
	PTGS2PDB_25244964_uff_E = 572.46	0	0	−8.5
	PTGS2PDB_25203368_uff_E = 541.72	0	0	−8.9
	PTGS2PDB_12795736_uff_E = 643.77	0	0	−8.8
	PTGS2PDB_11063337_uff_E = 244.73	0	0	−7.2
	PTGS2PDB_9828626_uff_E = 669.99	0	0	−8.8
	PTGS2PDB_9548595_uff_E = 638.65	0	0	−9.1
	PTGS2PDB_6419725_uff_E = 609.04	0	0	−8.6
	PTGS2PDB_5481663_uff_E = 826.93	0	0	−10.5
	PTGS2PDB_5318645_uff_E = 688.07	0	0	−9.2
	PTGS2PDB_5282761_uff_E = 74.80	0	0	−5.9
	PTGS2PDB_5281654_uff_E = 450.92	0	0	−8.9
	PTGS2PDB_5281243_uff_E = 655.10	0	0	−8.5
	PTGS2PDB_5281235_uff_E = 709.09	0	0	−8.4
	PTGS2PDB_5281230_uff_E = 422.22	0	0	−10.8
	PTGS2PDB_5281119_uff_E = 95.80	0	0	−6.6
	PTGS2PDB_5280934_uff_E = 142.21	0	0	−4.9
	PTGS2PDB_5280933_uff_E = 157.30	0	0	−5.7
	PTGS2PDB_5280899_uff_E = 704.15	0	0	−8.3
	PTGS2PDB_5280863_uff_E = 362.50	0	0	−9.1
	PTGS2PDB_5280805_uff_E = 751.59	0	0	−10.8
	PTGS2PDB_5280794_uff_E = 546.19	0	0	−8.5
	PTGS2PDB_5280791_uff_E = 556.51	0	0	−10.7
	PTGS2PDB_5280489_uff_E = 674.37	0	0	−10.8
	PTGS2PDB_5280450_uff_E = 147.99	0	0	−5.8
	PTGS2PDB_5280343_uff_E = 380.43	0	0	−8.2
	PTGS2PDB_689043_uff_E = 98.60	0	0	−6.9
	PTGS2PDB_638072_uff_E = 248.91	0	0	−5.9
	PTGS2PDB_637542_uff_E = 90.83	0	0	−6.5
	PTGS2PDB_637541_uff_E = 189.36	0	0	−6.2
	PTGS2PDB_637540_uff_E = 280.62	0	0	−6.6
	PTGS2PDB_493570_uff_E = 317.94	0	0	−7.8
	PTGS2PDB_445858_uff_E = 177.42	0	0	−7.1
	PTGS2PDB_445639_uff_E = 80.35	0	0	−6.8
	PTGS2PDB_445638_uff_E = 85.31	0	0	−5.4
	PTGS2PDB_445354_uff_E = 360.80	0	0	−6.8
	PTGS2PDB_444539_uff_E = 86.61	0	0	−6.5
	PTGS2PDB_444305_uff_E = 68.76	0	0	−5.3
	PTGS2PDB_241572_uff_E = 557.95	0	0	−8.6
	PTGS2PDB_222284_uff_E = 590.88	0	0	−8.8
	PTGS2PDB_173183_uff_E = 573.30	0	0	−9.8
	PTGS2PDB_145742_uff_E = 182.22	0	0	−5.0
	PTGS2PDB_101761_uff_E = 714.50	0	0	−8.9
	PTGS2PDB_101341_uff_E = 993.36	0	0	−8.5
	PTGS2PDB_94204_uff_E = 2069.23	0	0	−7.7
	PTGS2PDB_92110_uff_E = 2128.40	0	0	−8.3
	PTGS2PDB_73170_uff_E = 681.31	0	0	−9.4
	PTGS2PDB_73160_uff_E = 208.21	0	0	−8.6
	PTGS2PDB_65252_uff_E = 728.67	0	0	−8.6
	PTGS2PDB_33032_uff_E = 55.55	0	0	−5.1
	PTGS2PDB_14985_uff_E = 288.57	0	0	−7.4
	PTGS2PDB_13849_uff_E = 53.38	0	0	−5.4
	PTGS2PDB_11005_uff_E = 51.08	0	0	−5.2
	PTGS2PDB_10742_uff_E = 110.08	0	0	−6.1
	PTGS2PDB_10465_uff_E = 60.66	0	0	−6.6
	PTGS2PDB_8468_uff_E = 150.75	0	0	−6.4
	PTGS2PDB_7121_uff_E = 233.36	0	0	−6.3
	PTGS2PDB_6613_uff_E = 133.81	0	0	−6.6
	PTGS2PDB_6508_uff_E = 156.41	0	0	−6.5
	PTGS2PDB_6322_uff_E = 70.50	0	0	−5.9
	PTGS2PDB_6306_uff_E = 69.94	0	0	−4.9
	PTGS2PDB_6288_uff_E = 64.57	0	0	−4.6
	PTGS2PDB_6287_uff_E = 55.97	0	0	−4.6
	PTGS2PDB_6274_uff_E = 256.87	0	0	−5.4
	PTGS2PDB_6140_uff_E = 104.20	0	0	−6.2
	PTGS2PDB_6137_uff_E = 94.18	0	0	−4.5
	PTGS2PDB_6106_uff_E = 72.66	0	0	−5.1
	PTGS2PDB_6057_uff_E = 109.06	0	0	−6.7
	PTGS2PDB_5962_uff_E = 63.08	0	0	−4.8
	PTGS2PDB_5960_uff_E = 39.14	0	0	−4.9
	PTGS2PDB_5951_uff_E = 74.76	0	0	−4.5
	PTGS2PDB_5950_uff_E = 33.09	0	0	−4.2
	PTGS2PDB_5862_uff_E = 55.51	0	0	−4.0
	PTGS2PDB_4091_uff_E = 136.80	0	0	−5.3
	PTGS2PDB_3469_uff_E = 78.01	0	0	−6.3
	PTGS2PDB_2518_uff_E = 98.61	0	0	−7.0
	PTGS2PDB_1198_uff_E = 74.39	0	0	−5.8
	PTGS2PDB_1110_uff_E = 26.16	0	0	−4.8
	PTGS2PDB_1060_uff_E = 10.03	0	0	−4.5
	PTGS2PDB_1054_uff_E = 125.61	0	0	−5.5
	PTGS2PDB_971_uff_E = 8.52	0	0	−4.4
	PTGS2PDB_938_uff_E = 58.73	0	0	−5.4
	PTGS2PDB_750_uff_E = 27.51	0	0	−3.9
	PTGS2PDB_525_uff_E = 46.12	0	0	−5.2
	PTGS2PDB_370_uff_E = 77.82	0	0	−6.3
	PTGS2PDB_338_uff_E = 73.50	0	0	−5.9
	PTGS2PDB_311_uff_E = 79.82	0	0	−5.6
	PTGS2PDB_284_uff_E = 2.48	0	0	−2.8
	PTGS2PDB_176_uff_E = 4.30	0	0	−3.6

**Table 3 biology-14-00361-t003:** Molecular docking analysis of bioactive components from *Hippophae rhamnoides* L. with CCL2 protein: The tables present the binding affinity calculations between the bioactive compounds from *Hippophae rhamnoides* L. (sea buckthorn) and the target protein CCL2, which are identified as key druggable nodes in the diabetic cardiomyopathy network. The docking results show the potential interactions and binding strengths, indicating the therapeutic potential of sea buckthorn components in modulating these proteins and mitigating the effects of diabetes on cardiomyocytes.

Macromolecule	Phytochemical Compound PubChem ID	RMSD Upper Bound	RMSD Lower Bound	Molecular Docking: Binding Affinity (kcal/mol)
CCL2	CCL2PDB_157822370_uff_E = 101.29	0	0	−4.1
	CCL2PDB_54670067_uff_E = 200.65	0	0	−4.2
	CCL2PDB_25244964_uff_E = 572.46	0	0	−5.7
	CCL2PDB_25203368_uff_E = 541.72	0	0	−6.3
	CCL2PDB_12795736_uff_E = 643.77	0	0	−6
	CCL2PDB_11063337_uff_E = 244.73	0	0	−5
	CCL2PDB_9828626_uff_E = 669.99	0	0	−6.8
	CCL2PDB_9548595_uff_E = 638.65	0	0	−6.5
	CCL2PDB_6419725_uff_E = 609.04	0	0	−7
	CCL2PDB_5481663_uff_E = 826.93	0	0	−6.4
	CCL2PDB_5318645_uff_E = 688.07	0	0	−5.8
	CCL2PDB_5282761_uff_E = 74.80	0	0	−4.4
	CCL2PDB_5281654_uff_E = 450.92	0	0	−6
	CCL2PDB_5281243_uff_E = 655.10	0	0	−7.1
	CCL2PDB_5281235_uff_E = 709.09	0	0	−7.1
	CCL2PDB_5281230_uff_E = 422.22	0	0	−7.0
	CCL2PDB_5281119_uff_E = 95.80	0	0	−4.3
	CCL2PDB_5280934_uff_E = 142.21	0	0	−4.2
	CCL2PDB_5280933_uff_E = 157.30	0	0	−4.0
	CCL2PDB_5280899_uff_E = 704.15	0	0	−7.1
	CCL2PDB_5280863_uff_E = 362.50	0	0	−6.1
	CCL2PDB_5280805_uff_E = 751.59	0	0	−6.7
	CCL2PDB_5280794_uff_E = 546.19	0	0	−6.3
	CCL2PDB_5280791_uff_E = 556.51	0	0	−7.0
	CCL2PDB_5280489_uff_E = 674.37	0	0	−7.3
	CCL2PDB_5280450_uff_E = 147.99	0	0	−4.3
	CCL2PDB_5280343_uff_E = 380.43	0	0	−6.2
	CCL2PDB_689043_uff_E = 98.60	0	0	−4.8
	CCL2PDB_638072_uff_E = 248.91	0	0	−5.0
	CCL2PDB_637542_uff_E = 90.83	0	0	−4.7
	CCL2PDB_637541_uff_E = 189.36	0	0	−4.7
	CCL2PDB_637540_uff_E = 280.62	0	0	−5.1
	CCL2PDB_493570_uff_E = 317.94	0	0	−5.9
	CCL2PDB_445858_uff_E = 177.42	0	0	−4.7
	CCL2PDB_445639_uff_E = 80.35	0	0	−4.4
	CCL2PDB_445638_uff_E = 85.31	0	0	−3.9
	CCL2PDB_445354_uff_E = 360.80	0	0	−5.9
	CCL2PDB_444539_uff_E = 86.61	0	0	−4.8
	CCL2PDB_444305_uff_E = 68.76	0	0	−3.9
	CCL2PDB_241572_uff_E = 557.95	0	0	−5.9
	CCL2PDB_222284_uff_E = 590.88	0	0	−5.9
	CCL2PDB_173183_uff_E = 573.30	0	0	−6.0
	CCL2PDB_145742_uff_E = 182.22	0	0	−3.7
	CCL2PDB_101761_uff_E = 714.50	0	0	−6.5
	CCL2PDB_101341_uff_E = 993.36	0	0	−7.2
	CCL2PDB_94204_uff_E = 2069.23	0	0	−6.1
	CCL2PDB_92110_uff_E = 2128.40	0	0	−6.4
	CCL2PDB_73170_uff_E = 681.31	0	0	−7.1
	CCL2PDB_73160_uff_E = 208.21	0	0	−6.1
	CCL2PDB_65252_uff_E = 728.67	0	0	−6.6
	CCL2PDB_33032_uff_E = 55.55	0	0	−4.1
	CCL2PDB_14985_uff_E = 288.57	0	0	−4.7
	CCL2PDB_13849_uff_E = 53.38	0	0	−4.0
	CCL2PDB_11005_uff_E = 51.08	0	0	−3.7
	CCL2PDB_10742_uff_E = 110.08	0	0	−4.4
	CCL2PDB_10465_uff_E = 60.66	0	0	−3.7
	CCL2PDB_8468_uff_E = 150.75	0	0	−4.3
	CCL2PDB_7121_uff_E = 233.36	0	0	−4.5
	CCL2PDB_6613_uff_E = 133.81	0	0	−4.4
	CCL2PDB_6508_uff_E = 156.41	0	0	−4.5
	CCL2PDB_6322_uff_E = 70.50	0	0	−3.9
	CCL2PDB_6306_uff_E = 69.94	0	0	−3.6
	CCL2PDB_6288_uff_E = 64.57	0	0	−3.6
	CCL2PDB_6287_uff_E = 55.97	0	0	−3.5
	CCL2PDB_6274_uff_E = 256.87	0	0	−4.0
	CCL2PDB_6140_uff_E = 104.20	0	0	−4.6
	CCL2PDB_6137_uff_E = 94.18	0	0	−3.3
	CCL2PDB_6106_uff_E = 72.66	0	0	−3.8
	CCL2PDB_6057_uff_E = 109.06	0	0	−4.7
	CCL2PDB_5962_uff_E = 63.08	0	0	−3.5
	CCL2PDB_5960_uff_E = 39.14	0	0	−3.7
	CCL2PDB_5951_uff_E = 74.76	0	0	−3.2
	CCL2PDB_5950_uff_E = 33.09	0	0	−3.1
	CCL2PDB_5862_uff_E = 55.51	0	0	−3.1
	CCL2PDB_4091_uff_E = 136.80	0	0	−4.3
	CCL2PDB_3469_uff_E = 78.01	0	0	−4.8
	CCL2PDB_2518_uff_E = 98.61	0	0	−4.8
	CCL2PDB_1198_uff_E = 74.39	0	0	−4.2
	CCL2PDB_1110_uff_E = 26.16	0	0	−3.7
	CCL2PDB_1060_uff_E = 10.03	0	0	−3.3
	CCL2PDB_1054_uff_E = 125.61	0	0	−4.3
	CCL2PDB_971_uff_E = 8.52	0	0	−3.3
	CCL2PDB_938_uff_E = 58.73	0	0	−4.4
	CCL2PDB_750_uff_E = 27.51	0	0	−2.9
	CCL2PDB_525_uff_E = 46.12	0	0	−3.6
	CCL2PDB_370_uff_E = 77.82	0	0	−4.5
	CCL2PDB_338_uff_E = 73.50	0	0	−4.6
	CCL2PDB_311_uff_E = 79.82	0	0	−4.2
	CCL2PDB_284_uff_E = 2.48	0	0	−2.4
	CCL2PDB_176_uff_E = 4.30	0	0	−2.7

**Table 4 biology-14-00361-t004:** Molecular docking analysis of bioactive components from *Hippophae rhamnoides* L. with EGFR protein: The tables present the binding affinity calculations between the bioactive compounds from *Hippophae rhamnoides* L. (sea buckthorn) and the target protein EGFR, which are identified as key druggable nodes in the diabetic cardiomyopathy network. The docking results show the potential interactions and binding strengths, indicating the therapeutic potential of sea buckthorn components in modulating these proteins and mitigating the effects of diabetes on cardiomyocytes.

Macromolecule	Phytochemical Compound PubChem ID	RMSD Upper Bound	RMSD Down Bound	Molecular Docking: Binding Affinity (kcal/mol)
EGFR	EGFRPDB_157822370_uff_E = 101.29	0	0	−4.9
	EGFRPDB_54670067_uff_E = 200.65	0	0	−5.7
	EGFRPDB_25244964_uff_E = 572.46	0	0	−7.8
	EGFRPDB_25203368_uff_E = 541.72	0	0	−7.6
	EGFRPDB_12795736_uff_E = 643.77	0	0	−7.1
	EGFRPDB_11063337_uff_E = 244.73	0	0	−6
	EGFRPDB_9828626_uff_E = 669.99	0	0	−8.1
	EGFRPDB_9548595_uff_E = 638.65	0	0	−7.9
	EGFRPDB_6419725_uff_E = 609.04	0	0	−7.7
	EGFRPDB_5481663_uff_E = 826.93	0	0	−8
	EGFRPDB_5318645_uff_E = 688.07	0	0	−7.4
	EGFRPDB_5282761_uff_E = 74.80	0	0	−4
	EGFRPDB_5281654_uff_E = 450.92	0	0	−7.2
	EGFRPDB_5281243_uff_E = 655.10	0	0	−7.4
	EGFRPDB_5281235_uff_E = 709.09	0	0	−7.8
	EGFRPDB_5281230_uff_E = 422.22	0	0	−7
	EGFRPDB_5281119_uff_E = 95.80	0	0	−4.8
	EGFRPDB_5280934_uff_E = 142.21	0	0	−4.3
	EGFRPDB_5280933_uff_E = 157.30	0	0	−5.1
	EGFRPDB_5280899_uff_E = 704.15	0	0	−7.6
	EGFRPDB_5280863_uff_E = 362.50	0	0	−7.1
	EGFRPDB_5280805_uff_E = 751.59	0	0	−8.2
	EGFRPDB_5280794_uff_E = 546.19	0	0	−7.7
	EGFRPDB_5280791_uff_E = 556.51	0	0	−7.1
	EGFRPDB_5280489_uff_E = 674.37	0	0	−7.7
	EGFRPDB_5280450_uff_E = 147.99	0	0	−4.7
	EGFRPDB_5280343_uff_E = 380.43	0	0	−7.3
	EGFRPDB_689043_uff_E = 98.60	0	0	−5.7
	EGFRPDB_638072_uff_E = 248.91	0	0	−5.4
	EGFRPDB_637542_uff_E = 90.83	0	0	−5.9
	EGFRPDB_637541_uff_E = 189.36	0	0	−5.5
	EGFRPDB_637540_uff_E = 280.62	0	0	−5.3
	EGFRPDB_493570_uff_E = 317.94	0	0	−6.7
	EGFRPDB_445858_uff_E = 177.42	0	0	−6
	EGFRPDB_445639_uff_E = 80.35	0	0	−4.1
	EGFRPDB_445638_uff_E = 85.31	0	0	−4.4
	EGFRPDB_445354_uff_E = 360.80	0	0	−6.6
	EGFRPDB_444539_uff_E = 86.61	0	0	−5.8
	EGFRPDB_444305_uff_E = 68.76	0	0	−4.7
	EGFRPDB_241572_uff_E = 557.95	0	0	−7.2
	EGFRPDB_222284_uff_E = 590.88	0	0	−7.6
	EGFRPDB_173183_uff_E = 573.30	0	0	−7.7
	EGFRPDB_145742_uff_E = 182.22	0	0	−4.3
	EGFRPDB_101761_uff_E = 714.50	0	0	−8.1
	EGFRPDB_101341_uff_E = 993.36	0	0	−8.7
	EGFRPDB_94204_uff_E = 2069.23	0	0	−7.7
	EGFRPDB_92110_uff_E = 2128.40	0	0	−8
	EGFRPDB_73170_uff_E = 681.31	0	0	−8.1
	EGFRPDB_73160_uff_E = 208.21	0	0	−7.2
	EGFRPDB_65252_uff_E = 728.67	0	0	−7.1
	EGFRPDB_33032_uff_E = 55.55	0	0	−4.6
	EGFRPDB_14985_uff_E = 288.57	0	0	−5.6
	EGFRPDB_13849_uff_E = 53.38	0	0	−4.7
	EGFRPDB_11005_uff_E = 51.08	0	0	−4.4
	EGFRPDB_10742_uff_E = 110.08	0	0	−5.3
	EGFRPDB_10465_uff_E = 60.66	0	0	−4.3
	EGFRPDB_8468_uff_E = 150.75	0	0	−5.3
	EGFRPDB_7121_uff_E = 233.36	0	0	−5.3
	EGFRPDB_6613_uff_E = 133.81	0	0	−4.9
	EGFRPDB_6508_uff_E = 156.41	0	0	−5.7
	EGFRPDB_6322_uff_E = 70.50	0	0	−4.9
	EGFRPDB_6306_uff_E = 69.94	0	0	−4.3
	EGFRPDB_6288_uff_E = 64.57	0	0	−4.5
	EGFRPDB_6287_uff_E = 55.97	0	0	−4.3
	EGFRPDB_6274_uff_E = 256.87	0	0	−5.1
	EGFRPDB_6140_uff_E = 104.20	0	0	−5.8
	EGFRPDB_6137_uff_E = 94.18	0	0	−4
	EGFRPDB_6106_uff_E = 72.66	0	0	−4.6
	EGFRPDB_6057_uff_E = 109.06	0	0	−5.6
	EGFRPDB_5962_uff_E = 63.08	0	0	−4.3
	EGFRPDB_5960_uff_E = 39.14	0	0	−4.8
	EGFRPDB_5951_uff_E = 74.76	0	0	−4.2
	EGFRPDB_5950_uff_E = 33.09	0	0	−4
	EGFRPDB_5862_uff_E = 55.51	0	0	−3.8
	EGFRPDB_4091_uff_E = 136.80	0	0	−5
	EGFRPDB_3469_uff_E = 78.01	0	0	−5.6
	EGFRPDB_2518_uff_E = 98.61	0	0	−5.9
	EGFRPDB_1198_uff_E = 74.39	0	0	−5.4
	EGFRPDB_1110_uff_E = 26.16	0	0	−4
	EGFRPDB_1060_uff_E = 10.03	0	0	−3.8
	EGFRPDB_1054_uff_E = 125.61	0	0	−4.8
	EGFRPDB_971_uff_E = 8.52	0	0	−4
	EGFRPDB_938_uff_E = 58.73	0	0	−4.8
	EGFRPDB_750_uff_E = 27.51	0	0	−3.6
	EGFRPDB_525_uff_E = 46.12	0	0	−4.5
	EGFRPDB_370_uff_E = 77.82	0	0	−5.6
	EGFRPDB_338_uff_E = 73.50	0	0	−5.3
	EGFRPDB_311_uff_E = 79.82	0	0	−5.1
	EGFRPDB_284_uff_E = 2.48	0	0	−2.7
	EGFRPDB_176_uff_E = 4.30	0	0	−3.1

**Table 5 biology-14-00361-t005:** Molecular docking analysis of bioactive components from *Hippophae rhamnoides* L. with SERPINE1 protein: The tables present the binding affinity calculations between the bioactive compounds from *Hippophae rhamnoides* L. (sea buckthorn) and the target protein SERPINE1, which are identified as key druggable nodes in the diabetic cardiomyopathy network. The docking results show the potential interactions and binding strengths, indicating the therapeutic potential of sea buckthorn components in modulating these proteins and mitigating the effects of diabetes on cardiomyocytes.

Macromolecule	Phytochemical Compound PubChem ID	RMSD Upper Bound	RMSD Down Bound	Molecular Docking: Binding Affinity (kcal/mol)
SERPINE1	SERPINE1PDB_101341_uff_E = 993.36	0	0	−8.2
	SERPINE1PDB_101761_uff_E = 714.50	0	0	−8.3
	SERPINE1PDB_10465_uff_E = 60.66	0	0	−4.6
	SERPINE1PDB_1054_uff_E = 125.61	0	0	−6
	SERPINE1PDB_1060_uff_E = 10.03	0	0	−4
	SERPINE1PDB_10742_uff_E = 110.08	0	0	−5.7
	SERPINE1PDB_11005_uff_E = 51.08	0	0	−4.9
	SERPINE1PDB_11063337_uff_E = 244.73	0	0	−7.1
	SERPINE1PDB_1110_uff_E = 26.16	0	0	−4.6
	SERPINE1PDB_1198_uff_E = 74.39	0	0	−5.7
	SERPINE1PDB_12795736_uff_E = 643.77	0	0	−8
	SERPINE1PDB_13849_uff_E = 53.38	0	0	−5.1
	SERPINE1PDB_145742_uff_E = 182.22	0	0	−5.2
	SERPINE1PDB_14985_uff_E = 288.57	0	0	−6.8
	SERPINE1PDB_157822370_uff_E = 101.29	0	0	−5.2
	SERPINE1PDB_173183_uff_E = 573.30	0	0	−7.2
	SERPINE1PDB_176_uff_E = 4.30	0	0	−3.2
	SERPINE1PDB_222284_uff_E = 590.88	0	0	−8.3
	SERPINE1PDB_241572_uff_E = 557.95	0	0	−6.9
	SERPINE1PDB_2518_uff_E = 98.61	0	0	−6.7
	SERPINE1PDB_25203368_uff_E = 541.72	0	0	−7.9
	SERPINE1PDB_25244964_uff_E = 572.46	0	0	−9
	SERPINE1PDB_284_uff_E = 2.48	0	0	−3
	SERPINE1PDB_311_uff_E = 79.82	0	0	−5.4
	SERPINE1PDB_33032_uff_E = 55.55	0	0	−5.1
	SERPINE1PDB_338_uff_E = 73.50	0	0	−6
	SERPINE1PDB_3469_uff_E = 78.01	0	0	−5.8
	SERPINE1PDB_370_uff_E = 77.82	0	0	−6.5
	SERPINE1PDB_4091_uff_E = 136.80	0	0	−5.9
	SERPINE1PDB_444305_uff_E = 68.76	0	0	−5
	SERPINE1PDB_444539_uff_E = 86.61	0	0	−6.5
	SERPINE1PDB_445354_uff_E = 360.80	0	0	−6.4
	SERPINE1PDB_445638_uff_E = 85.31	0	0	−5.3
	SERPINE1PDB_445639_uff_E = 80.35	0	0	−5.2
	SERPINE1PDB_445858_uff_E = 177.42	0	0	−6
	SERPINE1PDB_493570_uff_E = 317.94	0	0	−8.8
	SERPINE1PDB_525_uff_E = 46.12	0	0	−5.1
	SERPINE1PDB_5280343_uff_E = 380.43	0	0	−8.2
	SERPINE1PDB_5280450_uff_E = 147.99	0	0	−5.1
	SERPINE1PDB_5280489_uff_E = 674.37	0	0	−7.9
	SERPINE1PDB_5280791_uff_E = 556.51	0	0	−6.7
	SERPINE1PDB_5280794_uff_E = 546.19	0	0	−8.6
	SERPINE1PDB_5280805_uff_E = 751.59	0	0	−8.2
	SERPINE1PDB_5280863_uff_E = 362.50	0	0	−7.8
	SERPINE1PDB_5280899_uff_E = 704.15	0	0	−7.3
	SERPINE1PDB_5280933_uff_E = 157.30	0	0	−5.2
	SERPINE1PDB_5280934_uff_E = 142.21	0	0	−5.6
	SERPINE1PDB_5281119_uff_E = 95.80	0	0	−5.4
	SERPINE1PDB_5281230_uff_E = 422.22	0	0	−6.9
	SERPINE1PDB_5281235_uff_E = 709.09	0	0	−7.9
	SERPINE1PDB_5281243_uff_E = 655.10	0	0	−7.1
	SERPINE1PDB_5281654_uff_E = 450.92	0	0	−8
	SERPINE1PDB_5282761_uff_E = 74.80	0	0	−4.7
	SERPINE1PDB_5318645_uff_E = 688.07	0	0	−7.6
	SERPINE1PDB_54670067_uff_E = 200.65	0	0	−5.9
	SERPINE1PDB_5481663_uff_E = 826.93	0	0	−8.3
	SERPINE1PDB_5862_uff_E = 55.51	0	0	−4.2
	SERPINE1PDB_5950_uff_E = 33.09	0	0	−4.4
	SERPINE1PDB_5951_uff_E = 74.76	0	0	−4.6
	SERPINE1PDB_5960_uff_E = 39.14	0	0	−4.5
	SERPINE1PDB_5962_uff_E = 63.08	0	0	−5
	SERPINE1PDB_6057_uff_E = 109.06	0	0	−6.1
	SERPINE1PDB_6106_uff_E = 72.66	0	0	−4.9
	SERPINE1PDB_6137_uff_E = 94.18	0	0	−4.6
	SERPINE1PDB_6140_uff_E = 104.20	0	0	−5.7
	SERPINE1PDB_6274_uff_E = 256.87	0	0	−5.2
	SERPINE1PDB_6287_uff_E = 55.97	0	0	−4.8
	SERPINE1PDB_6288_uff_E = 64.57	0	0	−4.7
	SERPINE1PDB_6306_uff_E = 69.94	0	0	−5.1
	SERPINE1PDB_6322_uff_E = 70.50	0	0	−5.6
	SERPINE1PDB_637540_uff_E = 280.62	0	0	−5.6
	SERPINE1PDB_637541_uff_E = 189.36	0	0	−5.8
	SERPINE1PDB_637542_uff_E = 90.83	0	0	−6.9
	SERPINE1PDB_638072_uff_E = 248.91	0	0	−5.3
	SERPINE1PDB_6419725_uff_E = 609.04	0	0	−7.7
	SERPINE1PDB_6508_uff_E = 156.41	0	0	−5.9
	SERPINE1PDB_65252_uff_E = 728.67	0	0	−8.1
	SERPINE1PDB_6613_uff_E = 133.81	0	0	−6.2
	SERPINE1PDB_689043_uff_E = 98.60	0	0	−6.9
	SERPINE1PDB_7121_uff_E = 233.36	0	0	−5.6
	SERPINE1PDB_73160_uff_E = 208.21	0	0	−7.5
	SERPINE1PDB_73170_uff_E = 681.31	0	0	−8.5
	SERPINE1PDB_750_uff_E = 27.51	0	0	−3.8
	SERPINE1PDB_8468_uff_E = 150.75	0	0	−6.1
	SERPINE1PDB_92110_uff_E = 2128.40	0	0	−7.7
	SERPINE1PDB_938_uff_E = 58.73	0	0	−5.7
	SERPINE1PDB_94204_uff_E = 2069.23	0	0	−8
	SERPINE1PDB_9548595_uff_E = 638.65	0	0	−7.4
	SERPINE1PDB_971_uff_E = 8.52	0	0	−4.2
	SERPINE1PDB_9828626_uff_E = 669.99	0	0	−7.9

## Data Availability

All of the raw data and the rest of the materials remain in the possession of Islamic Azad University—Isfahan (Khorasgan) Branch and are available upon request.

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
