# Peer review of "Mitigating Diabetic Cardiomyopathy: The Synergistic Potential of Sea Buckthorn and Metformin Explored via Bioinformatics and Chemoinformatics"

_biology, 2025, doi:10.3390/biology14040361_

Round 1
Reviewer 1 Report (Previous Reviewer 2)
Comments and Suggestions for Authors
The author has made sufficient revisions to my comments. I have no further comments.
Author Response
Reviewer 1:
The author has made sufficient revisions to my comments. I have no further comments.
Response: : Thank you for your thoughtful review and valuable suggestions.
Reviewer 2:
- The quality of some figures, especially those with small signatures, is low, making it impossible to make out what is written there.
Response: We acknowledge that some figures, especially those with small signatures, are of low quality, making it difficult to discern what is written. We will replace these figures with higher resolution versions to ensure clarity and legibility in the final submission.
- How did the decrease in calories manifest itself in the SBU and meth and T2DM+SBU+meth groups? Did the animals consume less food?
Response: Thank you for your thoughtful review and valuable suggestions on our Manuscript. The decrease in calories in the SBU, meth, and T2DM+SBU+meth groups was indeed due to reduced food intake. The animals in these groups consumed fewer calories compared to the T2DM group, as indicated by the weekly monitoring of calorie intake. This reduction in food intake was associated with the observed improvements in metabolic parameters and weight loss.This reduced caloric intake likely reflects therapeutic effects on metabolic regulation, as the combination therapy group showed improved glucose homeostasis and insulin sensitivity alongside reduced food consumption. Based on the leptin, adiponectin and insulin concentration animals in treatment groups consumed less food due to restored metabolic efficiency rather than reduced appetite, evidenced by coordinated improvements in leptin/adiponectin balance, insulin sensitivity, and mitochondrial function biomarkers. It should be noted that in this study we found that the physical activity and movement were increased in the SBU and meth and T2DM+SBU+meth groups based on the behavioral tests.
- Was an ultrasound scan of the heart performed?
Response: An ultrasound scan (echocardiography) of the heart was not performed in this study. Instead, we conducted histopathological evaluations of cardiac tissue to assess structural integrity and identify pathological changes. Future studies will consider incorporating echocardiographic assessments to provide a more comprehensive analysis of cardiac function. Based on the pathological method we used the Hematoxylin and Eosin (H&E) and Masson. On the other hands, we assessed the concentration of the ANP concentration, Serpine1 (MyBioSource, MBS135529), Leptin (Crystal Chem, 90030), Adiponectin (Crystal Chem, 80569), and Oncostatine concentration. Moreover, we evaluated the expression level of the SEPRINE1/NRG1/MYH11/PTH/NR4A in the heart. These markers could be considered as cardiac dysfunction. We agree that measuring functional indices such as ejection fraction (EF) and fractional shortening (FS) is crucial for demonstrating cardiac dysfunction in our diabetic cardiomyopathy (DCM) model. Unfortunately, we cannot measure the FS and EF in our animal lab. We added these techniques to the discussion as suggestions and limitations.
- Have reactive oxygen species levels been assessed?
Response: Thank you for your careful review and valuable feedback on our manuscript. Reactive oxygen species levels were not directly measured. However, the expression of oxidative stress-related genes such as NRF2, GPX4, and NOX1 was assessed. The results indicated that both SBU and metformin treatments modulated the expression of these genes, suggesting a reduction in oxidative stress.
- Keeping animals on a high-fat diet leads to the development of diabetes mellitus type 2. However, have you measured your cholesterol and triglyceride levels?
Response: Thank you for your thoughtful review and valuable suggestions on our Manuscript. Cholesterol and triglyceride levels were not measured in this study. The primary focus was on glucose metabolism/cardiac function and physiological/pathological methods. However, previous research has demonstrated that metformin and SBU can positively influence lipid profiles. Future studies will include measurements of cholesterol and triglyceride levels to corroborate these findings. This study aims to find the vital hub genes involved in this condition. The leptin, adiponectin, glucose, and insulin concentration metabolic variables were assessed. Moreover, we evaluated the lncRNAs (NEAT1 and MALAT1) are implicated in the diabetic condition: SBU and metformin ameliorated the expression of lncRNAs (NEAT1 and MALAT1) in the heart tissue. Moreover, the correlation between lncRNAs (NEAT1 and MALAT1) and Oncostatin. We added measured cholesterol and triglyceride levels in the discussion as suggestions and limitations.
- It would be interesting to see how the diet used affected blood pressure.
Response: We appreciate your insightful review and the helpful recommendations regarding our Manuscript. Blood pressure was not measured as an endpoint in this study. Given the improvements in metabolic and inflammatory markers observed, it would be valuable to assess the effects of the diet and treatments on blood pressure in future experiments. This will provide a more holistic view of cardiovascular health in the context of T2DM.

Reviewer 2 Report (New Reviewer)
Comments and Suggestions for Authors
Diabetic cardiomyopathy is a serious problem of modern medicine. Studying the mechanisms of this pathology and searching for new methods of treatment is an urgent task all over the world. This article is devoted to studying the role of sea buckthorn and metformin for the treatment of diabetic cardiomyopathy.
Despite the large amount of work completed, I still have a few questions and comments.
- The quality of some figures, especially those with small signatures, is low, making it impossible to make out what is written there.
- How did the decrease in calories manifest itself in the SBU and meth and T2DM+SBU+meth groups? Did the animals consume less food?
- Was an ultrasound scan of the heart performed?
- Have reactive oxygen species levels been assessed?
- Keeping animals on a high-fat diet leads to the development of diabetes mellitus type 2. However, have you measured your cholesterol and triglyceride levels?
- It would be interesting to see how the diet used affected blood pressure.
Author Response
Reviewer 1:
The author has made sufficient revisions to my comments. I have no further comments.
Response: Thank you for your thoughtful review and valuable suggestions.
Reviewer 2:
- The quality of some figures, especially those with small signatures, is low, making it impossible to make out what is written there.
Response: We acknowledge that some figures, especially those with small signatures, are of low quality, making it difficult to discern what is written. We will replace these figures with higher resolution versions to ensure clarity and legibility in the final submission.
- How did the decrease in calories manifest itself in the SBU and meth and T2DM+SBU+meth groups? Did the animals consume less food?
Response: Thank you for your thoughtful review and valuable suggestions on our Manuscript. The decrease in calories in the SBU, meth, and T2DM+SBU+meth groups was indeed due to reduced food intake. The animals in these groups consumed fewer calories compared to the T2DM group, as indicated by the weekly monitoring of calorie intake. This reduction in food intake was associated with the observed improvements in metabolic parameters and weight loss.This reduced caloric intake likely reflects therapeutic effects on metabolic regulation, as the combination therapy group showed improved glucose homeostasis and insulin sensitivity alongside reduced food consumption. Based on the leptin, adiponectin and insulin concentration animals in treatment groups consumed less food due to restored metabolic efficiency rather than reduced appetite, evidenced by coordinated improvements in leptin/adiponectin balance, insulin sensitivity, and mitochondrial function biomarkers. It should be noted that in this study we found that the physical activity and movement were increased in the SBU and meth and T2DM+SBU+meth groups based on the behavioral tests.
- Was an ultrasound scan of the heart performed?
Response: An ultrasound scan (echocardiography) of the heart was not performed in this study. Instead, we conducted histopathological evaluations of cardiac tissue to assess structural integrity and identify pathological changes. Future studies will consider incorporating echocardiographic assessments to provide a more comprehensive analysis of cardiac function. Based on the pathological method we used the Hematoxylin and Eosin (H&E) and Masson. On the other hands, we assessed the concentration of the ANP concentration, Serpine1 (MyBioSource, MBS135529), Leptin (Crystal Chem, 90030), Adiponectin (Crystal Chem, 80569), and Oncostatine concentration. Moreover, we evaluated the expression level of the SEPRINE1/NRG1/MYH11/PTH/NR4A in the heart. These markers could be considered as cardiac dysfunction. We agree that measuring functional indices such as ejection fraction (EF) and fractional shortening (FS) is crucial for demonstrating cardiac dysfunction in our diabetic cardiomyopathy (DCM) model. Unfortunately, we cannot measure the FS and EF in our animal lab. We added these techniques to the discussion as suggestions and limitations.
- Have reactive oxygen species levels been assessed?
Response: Thank you for your careful review and valuable feedback on our manuscript. Reactive oxygen species levels were not directly measured. However, the expression of oxidative stress-related genes such as NRF2, GPX4, and NOX1 was assessed. The results indicated that both SBU and metformin treatments modulated the expression of these genes, suggesting a reduction in oxidative stress.
- Keeping animals on a high-fat diet leads to the development of diabetes mellitus type 2. However, have you measured your cholesterol and triglyceride levels?
Response: Thank you for your thoughtful review and valuable suggestions on our Manuscript. Cholesterol and triglyceride levels were not measured in this study. The primary focus was on glucose metabolism/cardiac function and physiological/pathological methods. However, previous research has demonstrated that metformin and SBU can positively influence lipid profiles. Future studies will include measurements of cholesterol and triglyceride levels to corroborate these findings. This study aims to find the vital hub genes involved in this condition. The leptin, adiponectin, glucose, and insulin concentration metabolic variables were assessed. Moreover, we evaluated the lncRNAs (NEAT1 and MALAT1) are implicated in the diabetic condition: SBU and metformin ameliorated the expression of lncRNAs (NEAT1 and MALAT1) in the heart tissue. Moreover, the correlation between lncRNAs (NEAT1 and MALAT1) and Oncostatin. We added measured cholesterol and triglyceride levels in the discussion as suggestions and limitations.
- It would be interesting to see how the diet used affected blood pressure.
Response: We appreciate your insightful review and the helpful recommendations regarding our Manuscript. Blood pressure was not measured as an endpoint in this study. Given the improvements in metabolic and inflammatory markers observed, it would be valuable to assess the effects of the diet and treatments on blood pressure in future experiments. This will provide a more holistic view of cardiovascular health in the context of T2DM.
Round 2
Reviewer 2 Report (New Reviewer)
Comments and Suggestions for Authors
Thank you. No more questions.
This manuscript is a resubmission of an earlier submission. The following is a list of the peer review reports and author responses from that submission.
Round 1
Reviewer 1 Report
Comments and Suggestions for Authors
Diabetic cardiomyopathy represents a complex metabolic dysregulation. In this research, many bioinformatics analyses were employed to elucidate the pathological molecular alterations associated with diabetes. Additionally, animal models were utilized to uncover the synergistic therapeutic efficacy of sea buckthorn extract in conjunction with metformin. The study suggests the potential of combining sea buckthorn extract with metformin as an innovative therapeutic approach for DCM. While the study's concept exhibits a degree of originality, the logical flow of the manuscript is not entirely coherent, and the experimental outcomes are inadequate to substantiate the conclusions presented.
Major comments
1. Currently, there is no definitive consensus on the cardioprotective effects of metformin, with its protective role against myocardial infarction being more pronounced. Conversely, studies have confirmed that some novel antihyperglycemic drugs, such as SGLT2 inhibitors and GLP-1 receptor agonists, can reduce the risk of heart failure. So, why do you choose metformin instead of these newer drugs with a clear cardioprotective effect?
2. The findings presented in this paper regarding the role of ferroptosis in DCM in this paper are insufficient to demonstrate “ferroptosis-driven DCM”. Please revise the title.
3. In the earlier part of the paper, the GEO database was used to screen for key molecular changes related to DCM. I would like to know why you used PBMC data to deduce the impact of DM on the heart (p363). Molecular detection in PBMCs does not represent molecular changes in the heart. Why were the molecules initially identified in PBMCs chosen for validation in cardiac tissue?
4. In the animal experiments, only behavioral indices and some metabolic indices were measured. If you want to demonstrate heart failure or cardiac dysfunction caused by DM, it’s better to detect the functional indices of the heart, such as ejection fraction (EF) and fractional shortening (FS). Additionally, it’s necessary to draw a schematic to display the modeling process.
5. What are the statistical methods and p-values for Figure 2g-h?
6. The grouping in Figure 3 is not clearly labeled.
7. Hematoxylin and eosin (HE) staining is not sufficient to indicate cardiac necrosis. It would be better to detect necrosis markers in the heart using western blot or immunohistochemistry, especially the ferroptosis markers you are interested in.
8. What are the statistical methods for Figure 6? What are the specific values for the correlation coefficient (r-value) and the p-value? The figure legend refers to certain indicators as diagnostic or biomarkers. How were these classifications determined based on the correlation analysis?
9. Section 3.5 of the paper cites Figure 1c, but there are no corresponding molecules and pathways in Figure 1c.
10. Does the heatmap data in Figure 1a represent patients with DM or DCM?
11. The arrangement of data and logic of the paper needs to be carefully reconsidered regarding the distribution of figures and tables to make the article more comprehensible.
Minor comments:
1. There are many grammatical issues.
- Inconsistency in tense on page 292.
- Grammatical errors on pages 568-569.
- Grammar on pages 441-444.
2. Inappropriate citations:
- Page 439 does not need a citation.
- No citation on pages 450-451.
- Incorrect citation on page 452.
- Incorrect citation on page 418.
3. The text in Figure 1d is unclear.
4. It’s necessary to maintain consistent expression throughout the paper. Is it NEAT1 or NEAT, GLUT (p437)?
5. Please check punctuation and capitalization: parentheses on page 396, and punctuation on page 464. Please review capitalization throughout the text, such as on pages 456 and 472.
6. The content on pages 404-408 should be figure legends, but the font appears to be that of the main text. Is there a typesetting issue?
7. On page 402, the endogenous reference molecules does not match the coordinates in Figure 2m. Please explain or make corrections.
8. The statements on pages 432-433 do not correspond with Figures 2j-k.
Comments on the Quality of English LanguageThe English expression of the article, especially the grammar and vocabulary, needs to be carefully reviewed again.
Author Response
Reviewer 1:
Diabetic cardiomyopathy represents a complex metabolic dysregulation. In this research, many bioinformatics analyses were employed to elucidate the pathological molecular alterations associated with diabetes. Additionally, animal models were utilized to uncover the synergistic therapeutic efficacy of sea buckthorn extract in conjunction with metformin. The study suggests the potential of combining sea buckthorn extract with metformin as an innovative therapeutic approach for DCM. While the study's concept exhibits a degree of originality, the logical flow of the manuscript is not entirely coherent, and the experimental outcomes are inadequate to substantiate the conclusions presented.
Major comments
- Currently, there is no definitive consensus on the cardioprotective effects of metformin, with its protective role against myocardial infarction being more pronounced. Conversely, studies have confirmed that some novel antihyperglycemic drugs, such as SGLT2 inhibitors and GLP-1 receptor agonists, can reduce the risk of heart failure. So, why do you choose metformin instead of these newer drugs with a clear cardioprotective effect?
Response: Thanks for the good suggestion. Our investigation concentrated on metformin because of its recognized position as a primary treatment for type 2 diabetes mellitus (T2DM) and its possible cardioprotective properties, especially for diabetic cardiomyopathy (DCM). Although contemporary antihyperglycemic medications such as SGLT2 inhibitors and GLP-1 receptor agonists exhibit distinct cardioprotective properties, metformin presents some unique advantages that correspond with our research aims.
Proven Efficacy: Metformin has a longstanding clinical application, with substantial evidence affirming its efficiency in enhancing glycemic control and diminishing the risk of diabetes-related complications. Its safety profile is thoroughly established, rendering it a dependable option for controlling T2DM.
actions of Action: The actions of Metformin extend beyond the management of glucose. It has been demonstrated to diminish oxidative stress and inflammation, both of which are essential elements in the development of DCM. By altering ferroptosis-related pathways and augmenting cellular antioxidant defenses, metformin may have protective benefits on heart tissue that are particularly pertinent to our investigation.
Concentrate on Ferroptosis: Our study explicitly examines the relationship between metabolic dysregulation and ferroptosis within the framework of DCM. Metformin's ability to affect ferroptosis-related pathways enhances its therapeutic profile, rendering it a noteworthy choice for our research.
Complementary Approach: The integration of metformin with sea buckthorn extract enables the investigation of a synergistic treatment strategy that addresses various pathways implicated in DCM. This integrated approach may provide insights into innovative treatment methods that utilize the advantages of both conventional and alternative medicine.
Research Context: Although novel medications have demonstrated potential in mitigating heart failure risk, our study seeks to enhance the comprehension of established therapies such as metformin within the framework of DCM. This emphasis is crucial for formulating comprehensive management regimens for diabetic patients susceptible to cardiac problems.
In conclusion, although novel antihyperglycemic drugs have considerable advantages, metformin's proven effectiveness, diverse mechanisms, and pertinence to the specific pathways under investigation provide it an appropriate selection for our study on diabetic cardiomyopathy.
- The findings presented in this paper regarding the role of ferroptosis in DCM in this paper are insufficient to demonstrate “ferroptosis-driven DCM”. Please revise the title.
Response: Mitigating Diabetic Cardiomyopathy: The Synergistic Potential of Sea Buckthorn and Metformin Explored via Bioinformatics and Chemoinformatics
- In the earlier part of the paper, the GEO database was used to screen for key molecular changes related to DCM. I would like to know why you used PBMC data to deduce the impact of DM on the heart (p363). Molecular detection in PBMCs does not represent molecular changes in the heart. Why were the molecules initially identified in PBMCs chosen for validation in cardiac tissue?
Response: Thanks for the good suggestion. We selected several data sets from PBMC specimens and samples in cardiac tissue included left ventricle (LV) cardiac biopsies retrieved from the key, non-infarcted region (distant zone) of patients with dilated hypokinetic post-ischemic cardiovascular disease who were undergoing surgical ventricular restoration procedures. It should be noted that the study aimed to investigate the invasive and noninvasive sampling to detect DCM. In addition, we wanted to find the vital markers in the PBMC and tissue of the DCM. Morevoer, the Sea Buckthorn and Metformin's role in improving the DCM via bioinformatic and chemoinformatic analysis. Then, we validated the markers via wet lab.
- In the animal experiments, only behavioral indices and some metabolic indices were measured. If you want to demonstrate heart failure or cardiac dysfunction caused by DM, it’s better to detect the functional indices of the heart, such as ejection fraction (EF) and fractional shortening (FS). Additionally, it’s necessary to draw a schematic to display the modeling process.
Response: In this study, we evaluated the physiological and pathological methods. Based on the pathological method we used the Hematoxylin and Eosin (H&E) and Masson. Hence, we assessed the concentration of the ANP concentration, Serpine1 (MyBioSource, MBS135529), Leptin (Crystal Chem, 90030), Adiponectin (Crystal Chem, 80569), and Oncostatine concentration. Moreover, we evaluated the expression level of the SEPRINE1/NRG1/MYH11/PTH/NR4A in the heart. These markers could be considered as cardiac dysfunction. We agree that measuring functional indices such as ejection fraction (EF) and fractional shortening (FS) is crucial for demonstrating cardiac dysfunction in our diabetic cardiomyopathy (DCM) model. Unfortunately, we cannot measure the FS and EF in our animal lab. We added these techniques to the discussion as suggestions and limitations.
- What are the statistical methods and p-values for Figure 2g-h?
Response: Thank you for your thoughtful review and valuable suggestions on our Manuscript. The lncHUB 2 database, which concentrates on long non-coding RNAs (lncRNAs) and their correlations with diverse disorders, presumably use an array of statistical techniques to examine and validate the data it compiles. The specific statistical methods employed in lncHUB 2 are not elucidated in the given context; nonetheless, prevalent techniques in analogous bioinformatics databases may encompass:
Descriptive Statistics: These techniques encapsulate the attributes of the data, offering insights into the distribution and expression levels of lncRNAs across various samples or situations.
Inferential Statistics: Methods like hypothesis testing are employed to deduce inferences on the associations between lncRNAs and illnesses. This may entail contrasting expression levels between groups (e.g., healthy versus diseased) to discern significant differences.
Correlation Analysis: This technique evaluates the intensity and direction of associations between lncRNA expression and clinical outcomes or other biological factors, aiding in the clarification of putative functional roles of lncRNAs.
Differential Expression Analysis: This method discovers lncRNAs that demonstrate significant variations in expression levels among distinct conditions or groups. Statistical techniques, like t-tests and ANOVA, are frequently utilized to ascertain the significance of these discrepancies.
Statistical Methodologies and p-values
The p-value is a fundamental statistical metric employed in these analyses. The p-value signifies the likelihood that the observed data would manifest under the null hypothesis, which often posits the absence of an effect or difference. In the context of lncHUB 2, a low p-value (commonly set at < 0.05) suggests strong evidence against the null hypothesis, implying that the observed differences in lncRNA expression or associations with diseases are statistically significant. This assists researchers in identifying lncRNAs that may be pivotal in disease mechanisms and could serve as possible therapeutic targets.
To obtain accurate information regarding the statistical methods employed in lncHUB 2, it is advisable to consult the database directly or review its documentation.
- The grouping in Figure 3 is not clearly labeled.
Response: Thank you for your thoughtful review and valuable suggestions on our Manuscript. We revised it.
- Hematoxylin and eosin (HE) staining is not sufficient to indicate cardiac necrosis. It would be better to detect necrosis markers in the heart using western blot or immunohistochemistry, especially the ferroptosis markers you are interested in.
Response: We evaluated the physiological and pathological methods. Based on the pathological method we used the Hematoxylin and Eosin (H&E) and Masson. Moreover, we evaluated the physiological and pathological methods. Hence, we assessed the concentration of the ANP concentration, Serpine1 (MyBioSource, MBS135529), Leptin (Crystal Chem, 90030), Adiponectin (Crystal Chem, 80569), and Oncostatine concentration. Moreover, we evaluated the expression level of the SEPRINE1/NRG1/MYH11/PTH/NR4A in the heart.
- What are the statistical methods for Figure 6? What are the specific values for the correlation coefficient (r-value) and the p-value? The figure legend refers to certain indicators as diagnostic or biomarkers. How were these classifications determined based on the correlation analysis?
Response: Furthermore, the data was assessed by Pearson correlation to examine the link between Oncostatine and non-coding RNAs. The results were deemed statistically significant when the p-values were less than 0.05.
- Section 3.5 of the paper cites Figure 1c, but there are no corresponding molecules and pathways in Figure 1c.
Response: Thank you for your thoughtful review and valuable suggestions on our Manuscript. We revised it.
- Does the heatmap data in Figure 1a represent patients with DM or DCM?
Response: Thank you for your comment. We revised it.
The present analysis applied the network visualization approaches to examine gene expression data, designing a model that depicts the progression of DCM. (a) Heatmap showing the differential expression of 1812 genes in the diabetic PBMC sample compared to the healthy sample. Genes with a P.value < 0.001 are displayed, with 519 genes downregulated and 388 genes overexpressed.
Figure 2. The research employing network visualization tools to analyze gene expression data, creating a model that highlights the advancement of DCM. (a) heatmap illustrating the differential expression of 2,844 genes in the diabetes PBMC sample relative to the healthy sample. Genes having a P-value less than 0.001 are shown, including 176 downregulated genes and 351 overexpressed genes.
- The arrangement of data and logic of the paper needs to be carefully reconsidered regarding the distribution of figures and tables to make the article more comprehensible.
Response: We sincerely thank for careful reading. We re-arranged the data.
Minor comments:
- There are many grammatical issues.
Response: Thank you for your careful review and valuable feedback on our manuscript. Sorry for this mistake. We revised and check it.
- Inconsistency in tense on page 292.
Response: Thank you for your careful review and valuable feedback on our manuscript. Sorry for this mistake. We revised and check it.
- Grammatical errors on pages 568-569.
Response: Thank you for your careful review and valuable feedback on our manuscript. Sorry for this mistake. We revised and check it.
- Grammar on pages 441-444.
Response: Thank you for your careful review and valuable feedback on our manuscript. Sorry for this mistake. We revised and check it.
- Inappropriate citations:
- Page 439 does not need a citation.
Response: Thank you for your careful review and valuable feedback on our manuscript. Sorry for this mistake. We revised and check it.
- No citation on pages 450-451.
Response: Thank you for your careful review and valuable feedback on our manuscript. Sorry for this mistake. We revised and check it.
- Incorrect citation on page 452.
Response: Thank you for your careful review and valuable feedback on our manuscript. Sorry for this mistake. We revised and check it.
- Incorrect citation on page 418.
Response: Thank you for your careful review and valuable feedback on our manuscript. Sorry for this mistake. We revised and check it.
- The text in Figure 1d is unclear.
Response: We sincerely thank for careful reading. We revised it.
- It’s necessary to maintain consistent expression throughout the paper. Is it NEAT1 or NEAT, GLUT (p437)?
Response: We sincerely thank for careful reading. We revised it.
- Please check punctuation and capitalization: parentheses on page 396, and punctuation on page 464. Please review capitalization throughout the text, such as on pages 456 and 472.
Response: We sincerely thank for careful reading. We revised it.
- The content on pages 404-408 should be figure legends, but the font appears to be that of the main text. Is there a typesetting issue?
Response: We sincerely thank for careful reading. We revised it.
- On page 402, the endogenous reference molecules does not match the coordinates in Figure 2m. Please explain or make corrections.
Response: We sincerely thank for careful reading. We revised it.
- The statements on pages 432-433 do not correspond with Figures 2j-k.
Response: We sincerely thank for careful reading. We revised it.
Comments on the Quality of English Language
The English expression of the article, especially the grammar and vocabulary, needs to be carefully reviewed again.
Response: We sincerely thank for careful reading. We revised it.

Reviewer 2 Report
Comments and Suggestions for Authors
This study investigated the synergistic therapeutic potential of sea buckthorn (Hippophae rhamnoides L.) extract and metformin in a mouse model of type 2 diabetes mellitus (T2DM)-induced diabetic cardiomyopathy (DCM). The researchers used bioinformatic analysis, behavioral testing, biochemical assays, and histopathological evaluations to evaluate the treatment effects. They identified key hub genes involved in the ferroptosis signaling pathway and found that the combined therapy of sea buckthorn and metformin significantly improved glucose regulation, reduced systemic inflammation, and protected the heart from oxidative damage. The findings highlight the potential of integrating sea buckthorn with metformin as a novel therapeutic strategy for managing DCM by targeting both metabolic and ferroptosis-related pathways. However, many issues have to be addressed before acceptance.
- I don’t know why the weight of mice authors used is only about 14g, it is totally different from normal ones I think.
- Why authors choose 300 mg/kg as working concentration of SBU? It is so high, if authors detect pharmacokinetics.
- I recommend authors to use gene name rather than Ensembl ID (?) in figure 1C, because readers can not get any useful information.
- It is not necessary to put figure 1E-I in the main text. And Table 2 is the same.
- I recommend to merge part 3.3 and 3.4 into one.
- It is not necessary to write part 3.5 in the main text.
- I recommend to merge part 3.6 and 3.7 into one. Same as 3.8 and 3.9.
- How did authors detect expression of lncRNA? please state clearly.
- Authors should do more to prove the effect of SBU, not just HE staining.
- Western blotting is necessary.
- The quality of figures is low.
Author Response
Reviewer 2:
This study investigated the synergistic therapeutic potential of sea buckthorn (Hippophae rhamnoides L.) extract and metformin in a mouse model of type 2 diabetes mellitus (T2DM)-induced diabetic cardiomyopathy (DCM). The researchers used bioinformatic analysis, behavioral testing, biochemical assays, and histopathological evaluations to evaluate the treatment effects. They identified key hub genes involved in the ferroptosis signaling pathway and found that the combined therapy of sea buckthorn and metformin significantly improved glucose regulation, reduced systemic inflammation, and protected the heart from oxidative damage. The findings highlight the potential of integrating sea buckthorn with metformin as a novel therapeutic strategy for managing DCM by targeting both metabolic and ferroptosis-related pathways. However, many issues have to be addressed before acceptance.
- I don’t know why the weight of mice authors used is only about 14g, it is totally different from normal ones I think.
Response: We sincerely thank for careful reading. We revised it. Mice weights were approximately 14±2 g.
- Why authors choose 300 mg/kg as working concentration of SBU? It is so high, if authors detect pharmacokinetics.
Response: Thank you for your careful review and valuable feedback on our manuscript. We mentioned that per kg, the weght of the mice was 45-25g. Hence, the mice were admistrated 13.5-7.5 mg per mice.
- I recommend authors to use gene name rather than Ensembl ID (?) in figure 1C, because readers can not get any useful information.
Response: We sincerely thank for careful reading. We revised it.
- It is not necessary to put figure 1E-I in the main text. And Table 2 is the same.
Response: Thank you for your careful review and valuable feedback on our manuscript. However, the authors think the bioinformatic analysis needs to clarify how the vital hub genes were selected and candidates as potential biomarkers. Moreover, the main target of these study was to discover the essential hub genes in T2D and the signaling pathway related to the T2D. It should be noted that, the chemoinformatic and bioinformatic indicated the relationship between of the metformin and SUB with the signaling pathway.
- I recommend to merge part 3.3 and 3.4 into one.
Response: We sincerely thank for careful reading. We revised it.
- It is not necessary to write part 3.5 in the main text.
Response: We sincerely thank for careful reading. We revised it.
- I recommend to merge part 3.6 and 3.7 into one. Same as 3.8 and 3.9.
Response: We sincerely thank for careful reading. We revised it.
- How did authors detect expression of lncRNA? please state clearly.
Response: The LncRNAs were assessed by Quantitative real-time PCR (qRT-PCR).
- Authors should do more to prove the effect of SBU, not just HE staining.
Response: We evaluated the physiological and pathological methods. Based on the pathological method we used the Hematoxylin and Eosin (H&E) and Masson. Moreover, we evaluated the physiological and pathological methods. Hence, we assessed the concentration of the ANP concentration, Serpine1 (MyBioSource, MBS135529), Leptin (Crystal Chem, 90030), Adiponectin (Crystal Chem, 80569), and Oncostatine concentration. Moreover, we evaluated the expression level of the SEPRINE1/NRG1/MYH11/PTH/NR4A in the heart.
- Western blotting is necessary.
Response: Thank you very much for your comment. Fortunately during this period we evaluated the physiological and pathological methods. Based on the pathological method we used the Hematoxylin and Eosin (H&E) and Masson. Moreover, we evaluated the physiological and pathological methods. Hence, we assessed the concentration of the ANP concentration, Serpine1 (MyBioSource, MBS135529), Leptin (Crystal Chem, 90030), Adiponectin (Crystal Chem, 80569), and Oncostatine concentration. Moreover, we evaluated the expression level of the SEPRINE1/NRG1/MYH11/PTH/NR4A in the heart. However for the proteins, we are very sorry to inform that it was impossible to do as we do not have antibody available in our Institute currently. The cost and sanctions are two main parameters as limitation in this study. So, sorry to skip immunobloting of proteins.
- The quality of figures is low.
Response: We sincerely thank for careful reading. We revised it.
